# New high-resolution estimates of the permafrost thermal state and hydrothermal conditions over the Northern Hemisphere

Youhua Ran[1,3] *, Xin Li[2], Guodong Cheng[1,4], Jingxin Che[5], Juha Aalto[6,7], Olli Karjalainen[8], Jan Hjort[8], Miska Luoto[6], Huijun Jin[1,9], Jaroslav Obu[10], Masahiro Hori[11], Qihao Yu[1,3], Xiaoli Chang[12]

[1]Northwest Institute of Eco-Environment and Resources, Chinese Academy of Sciences, Lanzhou 730000, China.
[2]National Tibetan Plateau Data Center, State Key Laboratory of Tibetan Plateau Earth System, Environment and Resources , Institute of Tibetan Plateau Research, Chinese Academy of Sciences, Beijing 100101, China.
[3]University of Chinese Academy of Sciences, Beijing 100049, China.
[4]Institute of Urban Study, Shanghai Normal University, Shanghai 200234, China.
[5]School of Science, Nanchang Institute of Technology, Nanchang 330099, China.
[6]Department of Geosciences and Geography, University of Helsinki, P.O. Box 64, Gustaf Hällström in katu 2a, 00014 Helsinki, Finland.
[7]Finnish Meteorological Institute, P.O. Box 503, FI-00101 Helsinki, Finland.
[8]Geography Research Unit, University of Oulu, P.O. Box 8000, FI-90014, Oulu, Finland.
[9]Institute of Cold Regions Science and Engineering and School of Civil Engineering, Northeast Forestry University, Harbin 150040, China
[10]Department of Geosciences, University of Oslo, Postboks 1047 Blindern, 0316 Oslo, Norway.
[11]Earth Observation Research Center, Japan Aerospace Exploration Agency, 2-1-1, Sengen, Tsukuba, Ibaraki 305-8505, Japan.
[12]School of Resource & Environment and Safety Engineering, Hunan University of Science and Technology, Xiangtan 411201, China.

*Correspondence to*: Youhua Ran (ranyh@lzb.ac.cn)

**Abstract.** Monitoring the thermal state of permafrost (TSP) is important in many environmental science and engineering applications. However, such data are generally unavailable, mainly due to the lack of ground observations and the uncertainty of traditional physical models. This study produces novel permafrost datasets for the Northern Hemisphere (NH), including predictions of the mean annual ground temperature (MAGT) at the depth of zero annual amplitude (DZAA) (approximately 3 m to 25 m) and active layer thickness (ALT) with 1-km resolution for the period of 2000–2016, as well as estimates of the probability of permafrost occurrence and permafrost zonation based on hydrothermal conditions. These datasets integrate unprecedentedly large amounts of field data (1,002 boreholes for MAGT and 452 sites for ALT) and multisource geospatial data, especially remote sensing data, using statistical learning modelling with an ensemble strategy. Thus, the resulting data are more accurate than those of previous circumpolar maps (bias=0.02±0.16 °C, RMSE=1.32±0.13 °C for MAGT; bias=2.71±16.46 cm, RMSE=86.93±19.61 cm for ALT). The datasets suggest that the areal extent of permafrost (MAGT≤0 °C) in the NH, excluding glaciers and lakes, is approximately 14.77 (13.60–18.97) ×10$^6$ km$^2$ and that the areal extent of permafrost regions (permafrost probability>0) is approximately 19.82 ×10$^6$ km$^2$. The areal fractions of humid, semiarid/subhumid, and arid permafrost regions are 51.56%, 45.07%, and 3.37%, respectively. The areal fractions of cold (≤-3.0 °C), cool (-3.0 °C to -1.5 °C), and warm (>-1.5 °C) permafrost regions are 37.80%, 14.30%, and 47.90%, respectively. These new datasets based on

the most comprehensive field data to date contribute to an updated understanding of the thermal state and zonation of permafrost in the NH. The datasets are potentially useful for various fields, such as climatology, hydrology, ecology, agriculture, public health, and engineering planning. All of the datasets are published through the National Tibetan Plateau

Data Center (TPDC), and the link is https://data.tpdc.ac.cn/en/data/5093d9ff-a5fc-4f10-a53f-c01e7b781368 or https://doi.org/10.11888/Geocry.tpdc.271190 (Ran et al., 2021b).

## 1 Introduction

Permafrost is defined as ground that remains at or below 0 ℃ for at least two consecutive years (Van Everdingen, 2005). As a key component of the cryosphere in the Northern Hemisphere (NH), permafrost is sensitive to disturbances from climate

change, environmental disturbances and human activities (Shur and Jorgenson, 2007). Changes in the thermal state of permafrost (TSP) have multiple important impacts on infrastructure safety, hydrological cycles, ecosystems, and climate systems (Yoshikawa and Hinzman, 2003; Cheng, 2005; Cheng and Wu, 2007; Schuur et al., 2009; Tarnocai et al., 2009; Cheng and Jin, 2013; Gao et al., 2016; Hjort et al., 2018; Ran et al., 2018; Xie et al., 2019; Li et al., 2020a, 2020b).

Over the past half century, hundreds of permafrost maps have been compiled at local to global scales (e.g., Heginbottom et al.,

2002; Ran et al., 2012; Cao et al., 2019; Zhelezniak et al., 2021). On a global scale, the first permafrost map, the Circum-Arctic Map of Permafrost and Ground Ice Conditions, was compiled by the International Permafrost Association (IPA) using a manual delineation method by integrating all (readily) available data and regional permafrost maps in the 1990s (Heginbottom et al., 1993; Brown et al., 1997). The IPA permafrost map indicates that the permafrost region underlies an area spanning approximately $22.79 \times 10^6$ km$^2$ in the NH, while the estimated areal extent actually underlain by permafrost ranges

from $12.21 \times 10^6$ to $16.98 \times 10^6$ km$^2$ (Zhang et al., 2000, 2008). The IPA permafrost map represents the best understanding of the permafrost distribution in the NH in the 1990s and has been frequently used for model evaluation or validation and even to perform derivative permafrost simulations (e.g., Chadburn et al., 2017). However, its quality and consistency vary because it is derived from the experience and knowledge of experts and depends on very limited observations at varied scales and accuracies and limited integration methods of mapping.

Since the release of the IPA permafrost map, great advances have been made in monitoring and modeling permafrost. In terms of data, the amounts of both ground-based observations and remote sensing data have remarkably increased in the past 30 years. Global and regional observation networks have been gradually established worldwide, strengthening permafrost monitoring in many regions, including Russia (e.g., Dvornikov et al., 2016), North America (e.g., Romanovsky and Osterkamp, 2001; Smith et al., 2005, 2010), Central Asia and China (e.g., Zhao et al., 2010a, 2021), and Europe (e.g., Harris et al., 2001;

Mair et al., 2011; Kellerer-Pirklbauer, 2016). As a result of these regional networks, the Global Terrestrial Network for Permafrost (GTN-P) was established by the IPA in 1999 (Brown et al., 2000, 2008). The GTN-P monitors the TSP and active layer thickness (ALT) through the TSP and the Circumpolar Active Layer Monitoring (CALM) programs (Brown et al., 2008; Romanovsky et al., 2010; Biskaborn et al., 2015, 2019). At present, at a global scale, data on ground temperature measured at

approximately 700 boreholes and ALT data from more than 200 sites can be freely downloaded from the GTN-P website

(http://gtnp.arcticportal.org). The data at some monitoring sites have been accumulating for decades (e.g., Biskaborn et al., 2015, 2019; Luo et al., 2016). Remote sensing observations are unprecedentedly abundant, and some variables related to permafrost, such as land surface temperature (LST), vegetation cover, and snow cover, can be retrieved from remote sensing sensors with high accuracy and spatial-temporal resolution (e.g., Justice et al., 2002; Zhao et al., 2013). The most significant methodological advances in mapping permafrost mainly include the expansion of permafrost models to applications of various

spatial domains (Riseborough et al., 2008; Obu et al., 2019) and the extensive applications of machine learning to infer the occurrence and TSP (Aalto et al., 2018; Ran et al., 2021a).

With data accumulation and technical advances, several new hemispheric-scale permafrost maps have been compiled. Gruber (2012) proposed a simple semiempirical function relationship between the mean annual ground temperature (MAGT) and mean annual air temperature (MAAT) for estimating the global permafrost zonation at a 1-km scale using downscaled MAAT

data. The map indicates an areal extent of regions actually underlain by permafrost in the NH of approximately $12.9 \times 10^6$ to $17.7 \times 10^6$ km$^2$. Aalto et al. (2018) produced a distribution map of the circum-Arctic MAGT at the depth of zero annual amplitude (DZAA) using statistical forecasting models, with an estimated areal extent of regions actually underlain by permafrost of approximately $(15.1 \pm 2.8) \times 10^6$ km$^2$, excluding the areal extent of permafrost to the south of 30°N in the NH. The performance of statistical models depends heavily on the *in situ* MAGT data used as a training set and the adopted

predictors. The *in situ* data used in Aalto et al. (2018) were limited to those from the Qinghai-Tibet Plateau and Northeast China, while the predictor variables were mainly derived from downscaled monthly averages of climate data, which probably limited prediction accuracy from the perspective of permafrost mapping. Obu et al. (2019) employed an equilibrium state model for the temperature at the top of the permafrost (TTOP) for the 2000–2016 period, driven by remotely sensed LSTs, downscaled ERA-Interim climate reanalysis data, and a land cover map. The error of the modeled TTOP is ±2 °C in comparison

with the data obtained from permafrost boreholes. This discrepancy is probably due to the inherent differences in the TTOP and the MAGT (at the DZAA) and the shortcomings of the model structure, which lead to large uncertainties, especially in regions of warm (>−1 °C) permafrost, such as the Qinghai-Tibet Plateau (e.g., Wu et al., 2002; Riseborough, 2007; Zhao et al., 2017). The modelled areal extent of near-surface permafrost (TTOP<0 °C) covers an area of $13.9 \times 10^6$ km$^2$ (Obu et al., 2019). The European Space Agency (ESA) Climate Change Initiative (CCI) also provide permafrost products including MAGT,

ALT and permafrost probability that are derived from a remote sensing-driven CryoGrid model (Obu et al., 2021). The annual ALT and ground temperature at various depths during 1997-2019 with 1-km resolution are uniquely available for the permafrost and climate science communities. The abovementioned mapping efforts provide new generation permafrost maps. However, the accuracy of these maps is still limited, especially at middle and low latitudes and at high elevations, due to the limitations of the data and models. These limitations may be overcome by integrating more ground observations and remote

sensing data using statistical or machine learning models that can improve upon physical-based models by automatically identifying better solutions (Bergen et al., 2019).

On the other hand, the maps of both the IPA and Obu et al. (2019) present permafrost zonation based on areal continuity. Areal continuity-based map systems effectively reflect the permafrost distribution characteristics in high-latitude areas, but such systems are not suitable for describing high-altitude permafrost because the areal continuity of permafrost distribution is relative and scale-dependent (Nelson, 1989; Ran et al., 2012, 2021a). With a sufficiently high spatial resolution, all permafrost can be considered continuous or absent. Therefore, how to define the areal continuity of permafrost and the spatial resolution *per se* remain controversial and practically daunting. Additionally, continuity-based systems ignore the vertical distribution and longitudinal zonation of permafrost and thus cannot effectively reflect the hydrothermal conditions of permafrost, which are important for comprehensively understanding the characteristics and vulnerability of permafrost, especially at the regional scale (Cheng, 1984; Jin et al., 2014; Ran et al., 2021a). From a thermal stability perspective, for a given thermal condition and temperature increase in the air, permafrost temperature often responds more quickly in arid regions than in humid regions because of the much greater thermal inertia of wetter soils (Abu-Hamdeh, 2003). Dry soil can reduce evaporation heat consumption and increase the incident radiation on the soil (Pan and Mahrt, 1987). In addition, under different soil hydrological conditions, the response of permafrost to precipitation of various types may differ substantially (Trenberth and Shea, 2005). In humid regions, increased precipitation may heat the air and subsequently the active layer and permafrost, in contrast to that in arid regions. The difference in hydrothermal conditions is also reflected in the responses of ecosystems to precipitation because precipitation may alter or modify the effects of temperature on ecosystems (Zhao et al., 2018). Furthermore, ecosystems are very important for the thermal stability of permafrost under a changing climate (Shur and Jorgenson, 2007), although the interactions among climate, permafrost and ecosystems are complex.

In short, the current statement of permafrost maps requires addressing the need for more comprehensive and integrated efforts to map the thermal state and hydrothermal zonation of permafrost. Thus, the objectives of this paper are as follows: (1) to develop and release the MAGT and ALT datasets as a baseline at 1-km scale in the NH, (2) to provide revised zonal statistics of the thermal state and distribution of permafrost in the NH, and (3) to present a permafrost zonation map that features the detailed hydrothermal conditions of permafrost.

## 2 Materials and methods

We first compiled a ground measurement database for MAGT and ALT in the NH (see the workflow in Figure 1). Then, the MAGT and ALT predictions with a 1-km resolution in the NH were produced using ensemble statistical forecasting by integrating the remotely sensed freezing degree-days (FDD) and thawing degree-days (TDD) (i.e., the total annual degree-days below and above 0 ℃, respectively), leaf area index (LAI), snow cover duration (SCD), precipitation (mm), solar radiation (kJ m$^{-2}$ day$^{-1}$), soil organic content (g kg$^{-1}$), soil bulk density (kg m$^{-3}$), coarse fragment content (volumetric %), and the compiled ground measurement data. Correspondingly, a dataset for the probability of permafrost occurrence was also produced. Finally, the hydrothermal conditions of permafrost were produced by combining the MAGT and aridity index using a rule-based decision-making process.

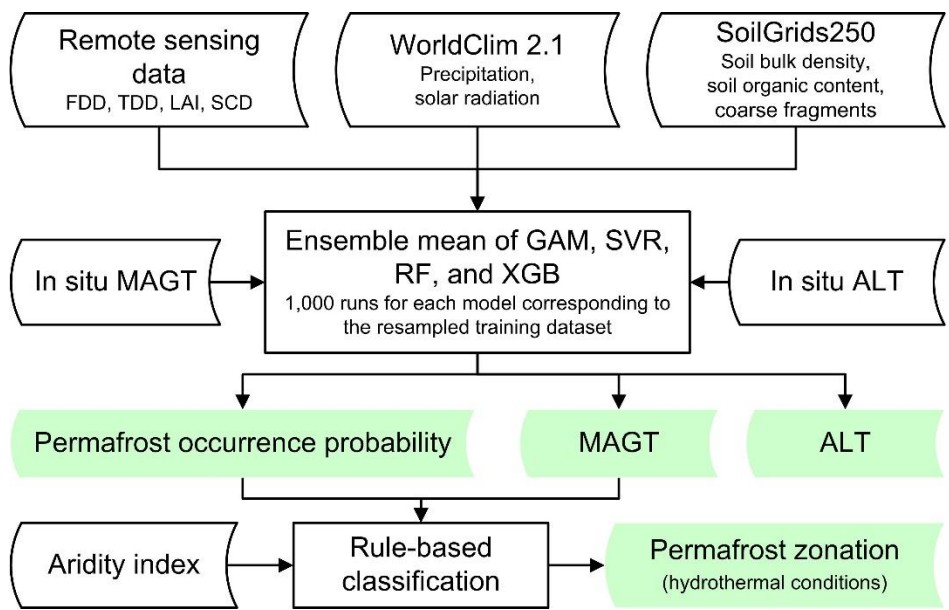

Figure 1: The data processing workflow used to compile the permafrost datasets (FDD: freezing degree-days, TDD: thawing degree-days, LAI: leaf area index, SCD: snow cover duration, MAGT: mean annual ground temperature, ALT: active layer thickness, GAM: generalized additive model, SVR: support vector regression, RF: random forest, and XGB: eXtreme gradient boosting).

## 2.1 Field measurement data

The standardized measurements of MAGT at or near DZAA (approximately 3 m to 25 m) from 1,002 boreholes and those of ALT from 452 sites were compiled mainly based on the ground measurement data used in Aalto et al. (2018), but the density of data points on the Qinghai-Tibet Plateau, in the Tianshan Mountains, and in Northeast China was greatly increased (Figure 2). Additional MAGT measurements from 253 boreholes are used mainly from the sources in Ran et al. (2021a), which were compiled mostly from existing literature on the Qinghai-Tibet Plateau (Wu et al., 2007; Yu et al., 2008; Sheng et al., 2010; Li et al., 2011, 2016; Zhang et al., 2011; Sun et al., 2013; Wang et al., 2013; Liu et al., 2015; Qiao et al., 2015; Wu et al., 2015; Qin et al., 2017; Cao et al., 2017; Luo, 2012; Luo et al., 2018a; Wani et al., 2020; Zhao et al., 2021), the Tianshan Mountains (Yu et al., 2013; Liu et al., 2015). Other additional MAGT measurements from 19 boreholes in Northeast China are from Li et al. (2019) and Chang (2011). Additional ALT measurements from 149 sites were mainly compiled from permafrost studies on the Qinghai-Tibet Plateau and the Tianshan Mountains (Zhao et al., 2010b; Luo et al., 2012, 2018b; Yu et al., 2013; Wu et al., 2015; Cao et al., 2017, 2018; Ali et al., 2018; Wani et al., 2020) and Northeast China (Chang, 2011; He et al., 2018; Li et al., 2020a). Only the measurement data from the undisturbed (natural) sites were used. Incomplete or inaccurate location information in some studies was corrected and checked carefully via communications with the authors. The elevations of these boreholes range from 0 m to 5,428 m above sea level (asl), and 99% of these measurements were made during 2000–2016. For MAGT, to reduce the potential overrepresentation of MAGT observations around 0 ℃, MAGT measurements in Aalto et al. (2018) both within and beyond the permafrost region were used (Karjalainen et al., 2019) but excluded the sites with MAGT>-15.5 ℃ (21 sites) to ensure the comparability of positive and negative temperature ranges, i.e., -15.5 ℃ and 15.5 ℃.

For some boreholes, exact DZAA values were not available, but multilayer time series measurement data were available; in these cases, DZAA was manually determined by examining the annual temperature variations (<0.1 ℃) at various depths assuming that averaged year-round temperature measurements can be used to derive more representative annual means than single measurements (Aalto et al., 2018; Karjalainen et al., 2019). For other boreholes where exact DZAA values were not

160    available but where single-time measurement data were available, the single-time temperature measurement at depths generally greater than 8 m (most approximately 15 m) below the ground surface were used, assuming that the depth was near the DZAA. Overall, the proportion of MAGTs at or near DZAA was quite large when we considered only permafrost sites (84%). However, for all 1002 sites, only 79% of MAGTs were at or near the DZAA (Table A1 in Appendix A). This was due to the inclusion of non-permafrost sites for which there often was no information to determine the DZAA.

165    To reduce the potential overfitting due to residual autocorrelation, we resampled the training data 1,000 times by excluding the sampling points within a distance of less than 3 km following Ran et al. (2021a). This resulted in using an average of 776 MAGT measurements and 276 ALT measurements for model training, and an average of 76 MAGT measurements and 30 ALT measurements were used for model evaluation per cross-validation run.

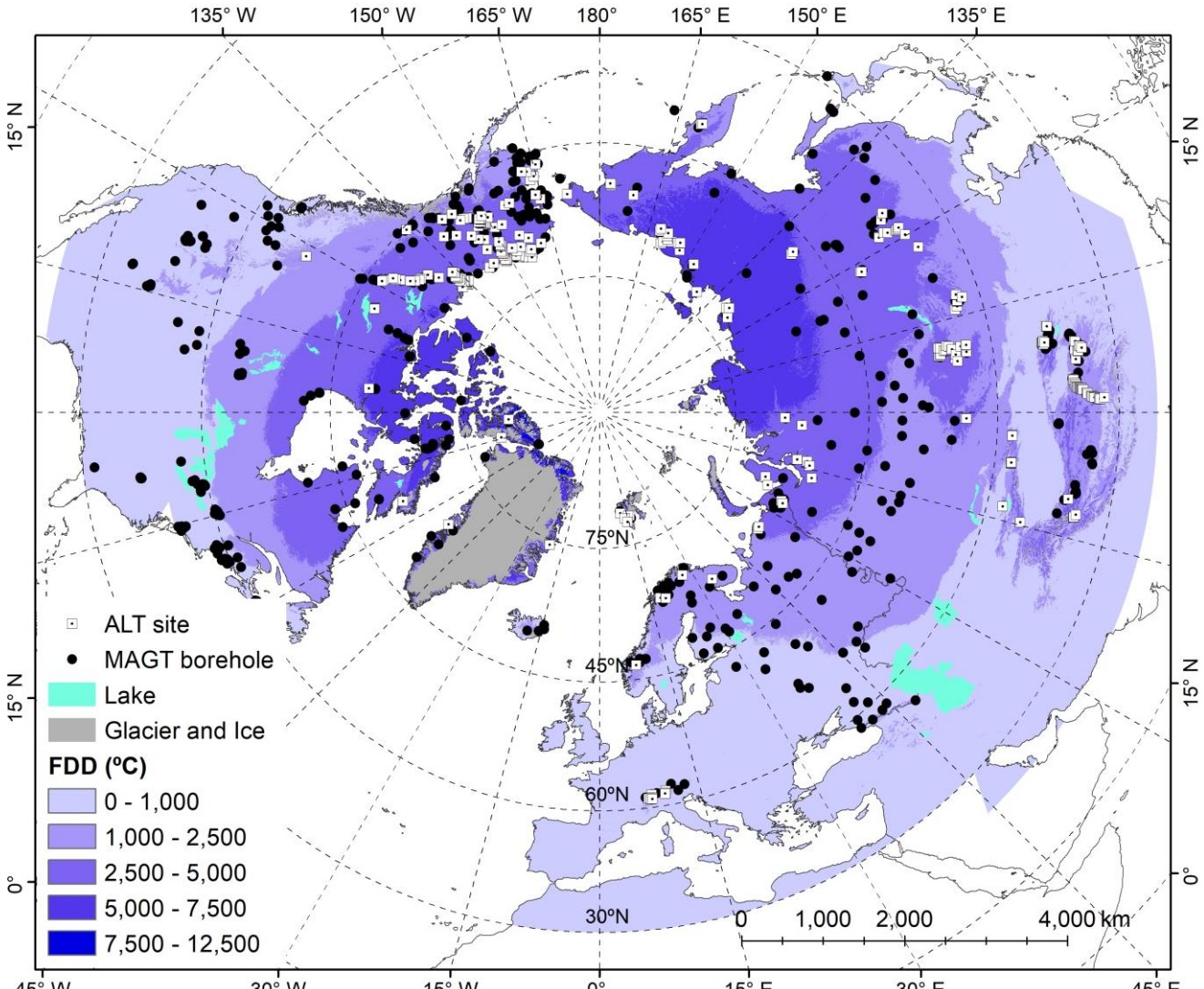

Figure 2: The distribution of the boreholes (1,002) for monitoring the mean annual ground temperature (MAGT) and the sites (452) for monitoring the active layer thickness (ALT) used in this study. FDD refers to the freezing degree-days ( ℃) derived from gap-filled daily MODIS land surface temperature.

## 2.2 Environmental and climate data

Nine environmental and climate variables were selected as predictors in a statistical learning model to estimate the permafrost thermal state based on previous studies (Aalto et al., 2018; Ran et al., 2021a). These variables were derived from high quality datasets available at present (Table 1).

Table 1: Environmental and climate variable datasets used in this study to predict MAGT (mean annual ground temperature) and ALT (active layer thickness). MODIS, Moderate Resolution Imaging Spectroradiometer; LST, land surface temperature; AVHRR, Advanced Very High Resolution Radiometer; GLASS, Global Land Surface Satellite

| Variable | Data source and ref. | Spatial resolution | Temporal resolution and time span |
|---|---|---|---|
| Freezing degree-days, ℃-days | MODIS LST, Obu et al., 2019 | 1 km | Daily, 2000-2016 |
| Thawing degree-days, ℃-days | MODIS LST, Obu et al., 2019 | 1 km | Daily, 2000-2016 |
| Snow cover duration, days | MODIS/AVHRR, Hori et al., 2017 | 0.05 ° | Half-month, 2000-2016 |
| Leaf area index | GLASS, Xiao et al., 2014 | 1 km | Eight-day, 2000-2016 |
| Precipitation, mm | WorldClim v2.1, Fick and Hijmans, 2017 | 1 km | 1970–2000 but adjusted to 2000–2016 |
| Solar radiation, kJ m$^{-2}$ day$^{-1}$ | WorldClim v2.1, Fick and Hijmans, 2017 | 1 km | 1970–2000 but adjusted to 2000–2016 |
| Soil organic content, g kg$^{-1}$ | SoilGrids250, Hengl et al., 2017 | 250 m | - |
| Soil bulk density, kg m$^{-3}$ | SoilGrids250, Hengl et al., 2017 | 250 m | - |
| Coarse fragment content, vol % | SoilGrids250, Hengl et al., 2017 | 250 m | - |

### 2.2.1 FDD and TDD

The multiannual average FDD and TDD with a spatial resolution of 1 km$^2$ were derived from four times daily Moderate Resolution Imaging Spectroradiometer (MODIS) LST (MOD11A1 and MYD11A1 version 6) from 2000 to 2016. Due to cloud contamination, gap-filling processing was applied to the MODIS LST time series using the downscaled ERA-Interim near-surface air temperature data (Dee et al., 2011; Obu et al., 2019) based on the assumption of the similarity of air and surface temperatures under cloudy skies (Gallo et al., 2011). More details about the gap-filling processing can be found in Obu et al. (2019).

### 2.2.2 Snow cover duration

The annual SCD data were estimated from satellite-derived bimonthly global snow cover extent (SCE) products obtained from the Japan Aerospace Exploration Agency (JAXA) Satellite Monitoring for Environmental Studies website. The SCE products with 0.05 °resolution were derived from polar-orbiting satellite-borne optical sensors (AVHRR and MODIS). Snow cover was identified using radiance data at five spectral bands from visible to thermal infrared wavelength regions same as those of AVHRR. The overall accuracy of snow/nonsnow cover classification was estimated to be 0.82–0.99. The detailed analysis method and accuracies of the SCE product can be found in Hori et al. (2017). The SCD is the annual sum of the temporal fraction of snow cover within each bimonthly period. Then, the multiannual average values of SCD from 2000 to 2016 were used in this study.

### 2.2.3 Leaf area index

The multiannual average values of LAI from 2000 to 2016 were derived from the Global Land Surface Satellite (GLASS), an eight-day, 1-km resolution LAI product. The GLASS LAI product was proposed based on the integration of MODIS and CYCLOPES LAI products, and the remaining cloud contamination values were removed using general regression neural networks (Xiao et al., 2014). The validation results show a higher accuracy for the GLASS LAI product in comparison with the MODIS and CYCLOPES LAI products (Xiang et al., 2014).

### 2.2.4 Soil data

The soil organic content, bulk density, and coarse fragment content source from SoilGrids250 (https://soilgrids.org) were used in this study. With a spatial resolution of 250 m, the SoilGrids250 is a global gridded soil map developed by the International Soil Reference and Information Center (ISRIC) using machine learning algorithms by integrating 230,000 soil profiles and remote sensing environmental variables (Hengl et al., 2017). The soil organic content, bulk density, and coarse fragment content were derived for seven standard depths (0, 5, 15, 30, 60, 100 and 200 cm) with a depth interval weighted average process and then aggregated to 30 arc sec resolution using the nearest resampling technique.

### 2.2.5 Downscaled climate data

The WorldClim v2.1 climate variables with a 1-km resolution (http://worldclim.org), including solar radiation (kJ m$^{-2}$ day$^{-1}$) and precipitation (mm) data (Fick and Hijmans, 2017), were used in this study. Solar radiation data were directly used, but the precipitation data for 1970–2000 were temporally adjusted to those for 2000–2016 based on WorldClim historical monthly weather data from 2000 to 2016 by using their locally smoothed (3×3 pixels) difference following Aalto et al. (2018).

### 2.3 Statistical learning model

The four statistical learning modelling techniques used in this study include the generalized additive model (GAM) (Hastie and Tibshirani, 1990), support vector regression (SVR) (Vapnik, 1995), random forest (RF) (Breiman, 2001), and eXtreme gradient boosting (XGB) (Friedman, 2001; Chen and Guestrin, 2016). The techniques were implemented based on the R packages mgcv (Wood, 2011) for GAM, randomForest (Liaw and Wiener, 2002) for RF, e1071 (Karatzoglou et al., 2006) for SVR, and xgboost (Chen and Guestrin, 2016) for XGB. The GAM is a semiparametric extension of the generalized linear model in which a smooth function is specified to fit the nonlinear function of multiple predictors to the response variable at the same time (Aalto et al., 2018). In this study, the maximum smoothing function was set to three, and the thin plate regression spline was used as the smoothing function. The RF is an ensemble learning algorithm for building and aggregating multiple decision trees. In this study, the number of trees is set to 400. Each tree is built using a randomly selected training dataset and three environmental variables to split each tree node. SVR features nonlinear kernel transformation, sparse solution, and maximal margin control (Awad and Khanna, 2015). It assumes that the maximum deviation ($\varepsilon$) (maximal margin) between the

predicted and measured values can be tolerated, and thus, an ε-insensitive loss function can be found to minimize the prediction error. The output model of the nonparametric technique depends on kernel functions. In this study, the default radial kernel function was used, and the grid search method based on 10-fold cross-validation was used to select the model parameters. A normalization method was used to avoid overfitting. XGB is an efficient implementation of a gradient boosted regression tree, which is an ensemble learning algorithm, and ensembles are built by merging multiple decision trees (Chen and Guestrin, 2016). The learning corrects the prediction errors from the prior models using an optimization algorithm by sequentially adding a tree. Here, XGB was implemented using the xgboost function through 50 rounds of 10-fold cross-validation, and max.depth was set as 3.

To account for the uncertainty of a single run and single model, an ensemble average of the 1,000 runs for the four model means was used to represent the distribution of MAGT and ALT in this study.

## 2.4 The probability of permafrost occurrence

The probability of permafrost occurrence is obtained by calculating the fraction of predicted MAGT≤0 °C based on the multimodel ensemble results with 1,000 runs. The areal extent of the permafrost region is defined as the regions where the probability of permafrost occurrence is greater than 0.

## 2.5 Hydrothermal condition-based permafrost zonation

We propose a classification system based on the hydrothermal condition of permafrost by synthesizing the thermal condition system proposed by Cheng (1984) and later modified by Ran et al. (2018) and by taking into account the aridity system proposed by Jin et al. (2014) and UNEP (1997). This system divides permafrost into nine categories using a two-level hierarchical index system, i.e., with two criteria consisting of the MAGT and climate aridity index (CAI). At the first level, permafrost is divided into cold (MAGT≤−3.0 ℃), cool (−3.0 ℃<MAGT≤−1.5 ℃), and warm (MAGT>−1.5 ℃) using MAGT as the sole indicator. At the second level, permafrost is divided into humid (CAI>0.65), semiarid/subhumid (0.65≥CAI>0.20), and arid (CAI≤0.20) using the CAI, the ratio of mean annual precipitation to potential evapotranspiration (P/PET). CAI data with 1-km resolution were used, sourced from the global aridity index database (Trabucco and Zomer, 2019) derived from precipitation data, and potential evapotranspiration was modelled using the Penman-Monteith equation based on WorldClim 2.1 climate data for the 1970–2000 period (http://worldclim.org/version2). Then, majority statistics processes with rectangular 5×5 neighbourhoods and boundary cleaning tools were used to remove the numerous small inclusions of permafrost zonation. Finally, glacier and lake areas were excluded. The extent of glaciers was sourced from the ESA CCI land cover map for 2010 (ESA, 2017), and the extent of lakes was sourced from the global lakes and wetlands database, level 1 (Lehner and Döll, 2004), which comprises large lakes (area ≥ 50 km$^2$) and large reservoirs (storage capacity ≥ 0.5 km$^3$).

## 2.6 Accuracy and uncertainty assessment

Model performance was assessed on the basis of the root-mean-square error (RMSE), bias, and square of the correlation coefficient ($R^2$) computed by distance-blocked 10-fold cross-validation with 1,000 repetitions. The uncertainty of the permafrost area was quantified using the 97.5th and 2.5th percentiles of the multimodel ensemble-simulated MAGT with 1,000 runs.

## 3 Results and discussion

The cross-validation indicates that the ensemble average of four statistical techniques (GAM, SVR, RF, and XGB) achieved the highest accuracy for MAGT (RMSE=1.32±0.13 ℃, bias=0.02±0.16 ℃), but the RF model was significantly more accurate (RMSE=85.47±20.39 cm, bias=-1.01±16.49 cm) than the ensemble mean (RMSE=86.93±19.61 cm, bias=2.71±16.46 cm) for ALT prediction (p≤0.001, paired sample t test, n=1,000) (Table 2). The accuracy of the predicted MAGT in permafrost regions, with field measurements in permafrost sites (MAGT ≤0 ℃) used as reference, was significantly higher (RMSE=1.06 ℃, bias=-0.22 ℃) than that in non-permafrost regions (RMSE=1.56 ℃, bias=0.88 ℃).

Based on the datasets (named NIEER in reference to the Northwest Institute of Eco-Environment and Resources, Chinese Academy of Sciences), we analysed the distribution characteristics of MAGT, ALT, and the permafrost area/region changes along latitude, elevation, and aridity index transects and the hydrothermal conditions in the permafrost regions.

**Table 2: The predictive performance (mean±SD) of the mean annual ground temperature (MAGT) and active layer thickness (ALT) for the four statistical learning models and their ensemble means.**

| Variable | Performance measures | GAM | SVR | RF | XGB | Ensemble average |
|---|---|---|---|---|---|---|
| MAGT | RMSE ( ℃) | 1.46±0.13 | 1.40±0.15 | 1.40±0.14 | 1.44±0.15 | 1.32±0.13 |
| | Bias ( ℃) | 0.01±0.18 | 0.04±0.18 | 0.02±0.17 | 0.03±0.18 | 0.02±0.16 |
| | $R^2$ | 0.97±0.01 | 0.97±0.01 | 0.97±0.01 | 0.97±0.01 | 0.97±0.01 |
| ALT | RMSE (cm) | 99.40±32.91 | 89.20±19.94 | 85.47±20.39 | 91.76±21.24 | 86.93±19.61 |
| | Bias (cm) | −0.63±19.35 | 12.09±16.85 | −1.01±16.49 | 0.41±17.24 | 2.71±16.46 |
| | $R^2$ | 0.72±0.14 | 0.76±0.08 | 0.78±0.08 | 0.75±0.09 | 0.77±0.08 |

Notes: GAM=generalized additive model, SVR=support vector regression, RF=random forest, and XGB=eXtreme gradient boosting.

## 3.1 Mean annual ground temperature and active layer thickness in the NH

Figure 3 shows the distribution of MAGT in the NH displaying an obvious latitudinal gradient from the zone of extremely cold ($<-10$ ℃) permafrost in the High Arctic to the zone of warm ($>-2$ ℃) permafrost in alpine and high-plateau regions at low latitudes, such as the Qinghai-Tibet and Mongolian plateaus, as well as the Yablonovy and Stanovoy mountains in southeastern Russia. In the Arctic permafrost region, the MAGT shows a clear distribution pattern of Arctic mountain permafrost in the Eastern Siberian Lowlands, on the Central Siberian Plateau, and in the Ural Mountains, Scandinavia in Europe, and the upper Yukon River Basin in western Canada. The average MAGT is approximately $-1.56\pm1.06$ ℃ in the Third Pole, which is cored by the Qinghai-Tibet Plateau, while that in the Arctic is approximately $-4.70\pm3.13$ ℃; the average MAGT is especially cold in the High Arctic, where it is close to $-9.5$ ℃. The pattern of ALT in the NH is generally similar to that of MAGT, but the details vary markedly (Figure 4). The regional average ALT varies from $76.95\pm21.69$ cm in the High Arctic to $232.40\pm47.95$ cm in the alpine and high-plateau permafrost regions at low latitudes with a narrow transition zone in the Mongolian Plateau and Northeast China.

The three-dimensional ground thermal regimes across the NH are investigated on the basis of the simulated MAGT and ALT. Figure 5a illustrates the latitudinal distributive patterns of MAGT and ALT, which show some comparable and contrasting features between those of MAGT and ALT. Regional average MAGT is generally stable at approximately $-1.5$ ℃ from 28 ℃N in the Himalayas to 60 ℃N in the Subarctic with a stable standard deviation. From 60 ℃N northwards, with rising latitudes, the MAGT decreases linearly, but its variation increases. In contrast, the regional average ALT is nearly stable at 80 cm between 56° and 84 ℃N, and there is a narrow transitional zone between 56 ℃N and 46 ℃N, where the ALT enlarges rapidly from approximately 100 to 250 cm. These trends are considered to be predominantly associated with the transition from alpine/high-elevation plateau permafrost to lowland permafrost with generally more protective ecosystem properties (e.g., thicker overburdens). The contrasting latitudinal zonation in MAGT and ALT may indicate their varied geoenvironmental impacts on MAGT and ALT at different scales. The ALT represents the hydrothermal state near the ground surface with more spatiotemporal heterogeneity than the MAGT, which represents the thermal state of the relatively deeper ground. The vulnerability of the near-surface ground to external disturbances associated with the inconsistency of the ALT measurement method may be one of the reasons for the large uncertainty in the prediction of the ALT. Of course, the uncertainty of ALT is considerable, especially in the vast area of western Siberia where the training data are sparse. The low spatial representativeness of training data may lead to an overestimation in several Siberian mountain regions and underestimation near the lower boundary of permafrost. This highlights the importance and urgency of strengthening global coordinated ALT observation networks. Overall, the complexity of predictive models of ALT needs further investigation.

The elevational effects on MAGT and ALT are modified by latitudinal effects at the hemispheric scale because mountains are widely distributed in the NH, with more in the lower latitudes. The combined effect of latitude and elevation differs among different elevation ranges. From 0 to 2,500 m asl, contrary to our understanding of the elevational effect at the regional scale, MAGT and ALT increase with increasing elevation, with large variations (Figure 5b). However, the variations show opposite

trends within this elevation range: with increasing elevation, the variation in MAGT declines, whereas that in ALT increases. In this elevation range, the elevational effect may be completely dominated by the latitudinal effect. The combined effect of latitude and elevation is moderate between 2,500 and 5,000 m asl, where MAGT is stable at approximately −2 °C, with minor variation, and ALT is stable at approximately 220 cm. At elevations >5,000 m asl, the effect of elevation appears. MAGT and ALT both decline monotonically with increasing elevation.

The trend of MAGT corresponds well to that of ALT along the rising aridity index (Figure 5c). For CAI values from 0.2 to 0.5, the MAGT and ALT both rapidly decrease with increasing CAI (moisture). However, in the range of CAI values from 0.5 to 3.6, MAGT and ALT both increase with increasing CAI. Then, above values of 3.6 on the CAI, both MAGT and ALT are nearly stable or slightly decreasing.

The three-dimensional zonal (latitudinal, elevation, and aridity) characteristics of MAGT and ALT discussed above probably indicate different controlling factors for MAGT and ALT and their variations. Multilinear regression analysis shows that the contributions of precipitation and soil bulk density to MAGT are statistically significant ($p<0.01$), but those to ALT are insignificant, while the contribution of the coarse fragment content of soils to MAGT is significant ($p<0.01$), whereas that to ALT is insignificant. The FDD, TDD, SCD, LAI, soil organic content, and solar radiation variables all contribute significantly to both MAGT and ALT ($p<0.05$). These differences may also be affected by the data gaps and uncertainty and analysis scales. In general, at the hemisphere scale, latitude dominates the distributive patterns and trends of MAGT and ALT. However, elevation and aridity become the main factors at the regional scale. For example, in mountainous areas of sub-Arctic and low-latitude permafrost regions, elevation becomes the controller, especially above 5,000 m asl (Figure 5b). In plateau areas, such as the Qinghai-Tibet Plateau, which are dominated by semiarid/subhumid and arid hydrologic conditions, aridity becomes the main controller of ALT ($p<0.05$). This role of aridity has not received due attention in the literature, and further research is very much needed. This highlights the necessity of the preliminary hydrothermal zonation of permafrost in Section 3.3.

In addition, the DZAA represented by predicted MAGT varies in the NH (approximately 3 m to 25 m). In general, in the continuous permafrost region in the Arctic and Qinghai-Tibet Plateau, the range in annual air temperature is large, and the ZAA is reached only at depths of 10 to 25 m below the surface (Ran et al., 2021a). These depths are greater than those in discontinuous permafrost and midlatitude regions, where the DZAA occurs at 5-10 m or less. This pattern is notably different from those of other MAGT products for specific depths.

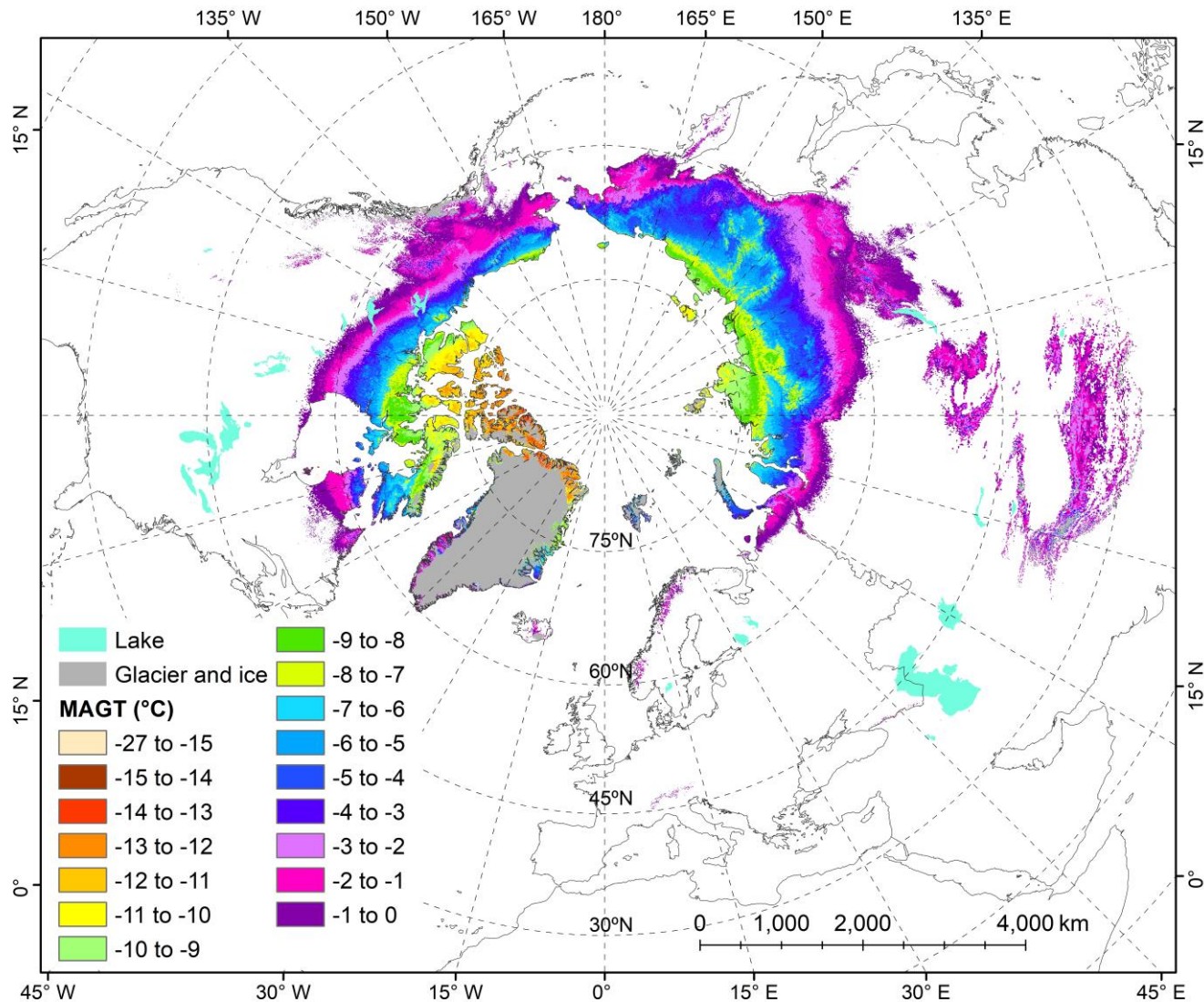

**Figure 3: Distribution of the mean annual ground temperature (MAGT) at the zero annual amplitude depth (approximately 3 m to 25 m) average in the Northern Hemisphere for the period of 2000-2016.**


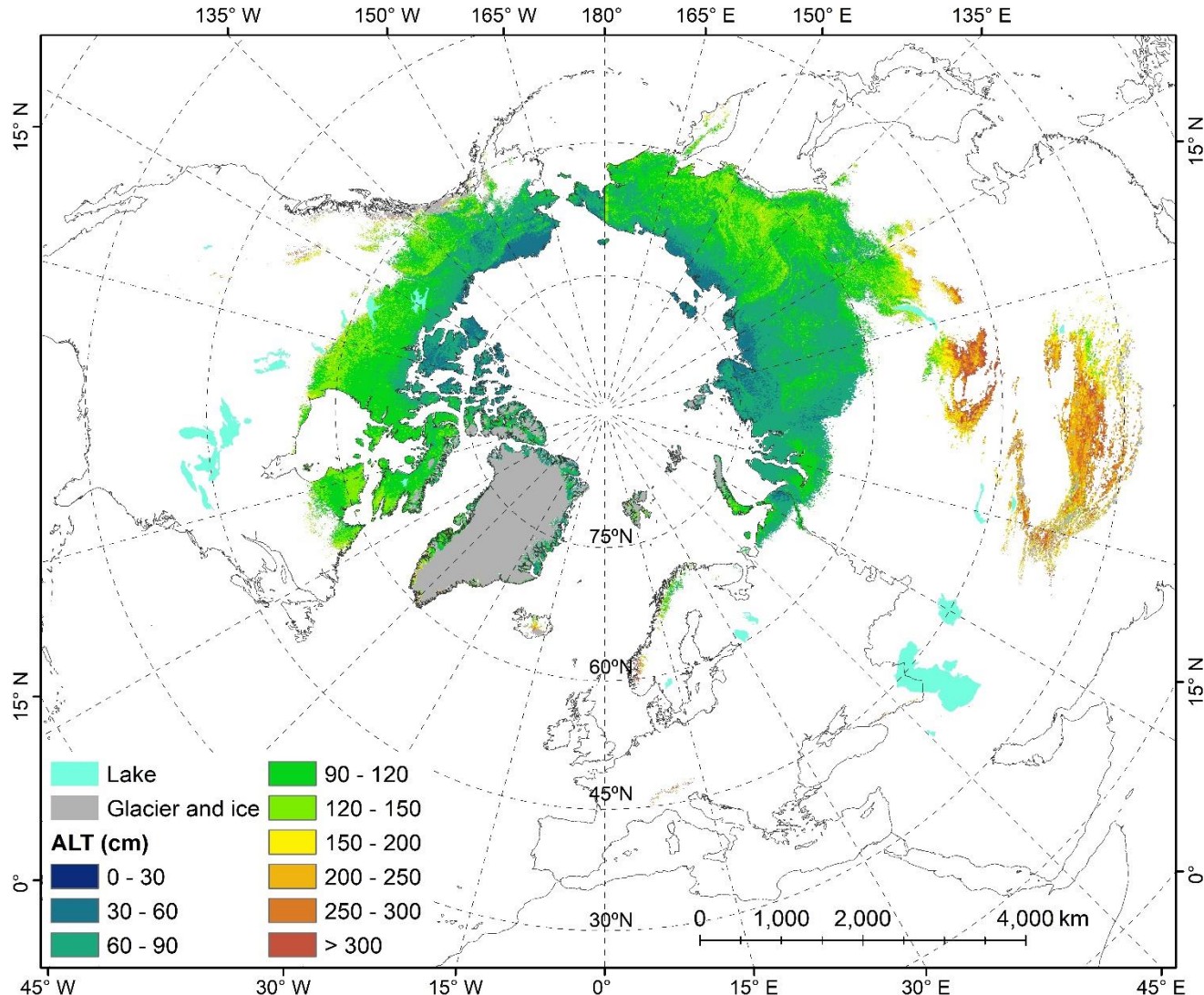

**Figure 4: Distribution of the average active layer thickness (ALT) in the Northern Hemisphere for the period of 2000-2016.**

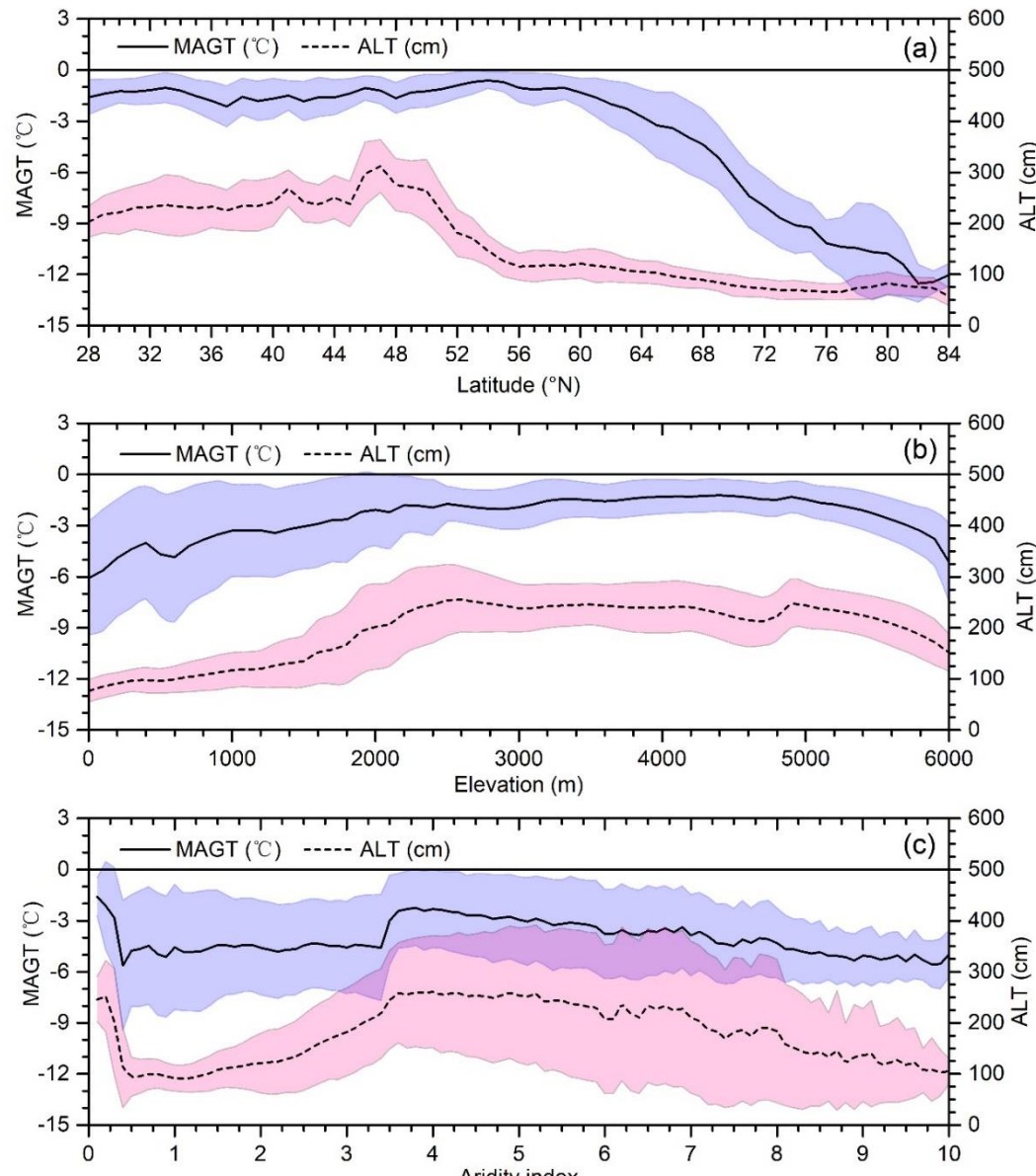

**Figure 5: Mean annual ground temperature (MAGT) and active layer thickness (ALT) along latitude (a), elevation (b), and aridity index (P/PET) (c) transects in the Northern Hemisphere (The black solid line is mean MAGT, and the light blue shaded area is its standard deviation. The dashed line is mean ALT, and the pink shaded area is its standard deviation. The dark pink area in (c) is the overlapped shaded area between ALT and MAGT).**

### 3.2 Permafrost distribution in the NH

For permafrost distribution, two terms, i.e., permafrost area and permafrost region, should be distinguished. In general, the permafrost area is defined as the area with MAGT≤0 °C, while the permafrost region is defined as the area where the permafrost

probability is greater than 0 (Zhang et al., 2000). According to the ensemble average of MAGT, the permafrost area over the NH, excluding glaciers (approximately $0.64 \times 10^6 \, km^2$) and lakes (approximately $0.3 \times 10^6 \, km^2$), is approximately 14.77 (13.60–18.97) $\times 10^6 \, km^2$. The range in permafrost area was quantified by using the 97.5th and 2.5th percentiles of the multimodel ensemble-simulated MAGT with 1,000 runs. This result is generally consistent with those of existing studies (Zhang et al.,

2008; Gruber, 2012; Chadburn et al., 2017; Aalto et al., 2018; Obu et al., 2019) and logically consistent with the ESA CCI permafrost area defined according to MAGT at 2 m (Obu et al., 2021). Our results contain permafrost in discontinuous zones while ESA CCI data contain no permafrost, because permafrost still may occur in deeper layers but has thawed at shallower depths. This difference is substantial particularly in the pan-Arctic. The new data indicate that the areal extent of the permafrost region is approximately $19.82 \times 10^6 \, km^2$ with different occurrence probabilities (Figure 6). This area is slightly less

than that estimated in a recent report (Obu et al., 2019) and approximately $2.59 \times 10^6 \, km^2$ less than that estimated in the IPA Circum-Arctic permafrost map under the same conditions (i.e., excluding glaciers and water bodies). The differences in permafrost region between the NIEER map proposed in this study and the IPA map are distributed mainly in the southern/lower boundary of permafrost, such as in the Mongolian Plateau and Qinghai-Tibet Plateau (Figure 7). This difference mainly reflects the difference in cartography technology between the two eras. The IPA map is a mechanical compilation of national maps,

with each nation having their own mapping standard, which introduces multiple errors and uncertainties. Furthermore, the permafrost region of the new map uses permafrost probability (>0), based on multimodel ensemble-simulated MAGT (<=0 ℃), as the boundary. This probability mainly reflects the spatial variation of uncertainty of the ensemble model and differs from the definition of permafrost occurrence probability in Gruber (2012) and Cao et al. (2019). In the real world, MAGT (<=0 ℃) is a very strict standard due to the effect of thermal offset within the active layer, and permafrost may exist between the DZAA

and the depth of seasonal maximum thaw (i.e., the ALT).

In general, at a global scale, latitude, elevation and aridity/longitude mainly govern the distribution of permafrost (Cheng, 1984; Noetzli et al., 2017). The latitudinal dependence of the permafrost distribution is based on the latitudinal variation in insolation (incoming solar radiation) and surface energy balance, and thus on the subsequent latitudinal zonation of climate, soil, and vegetation. The dependence of permafrost distribution on elevation is due to the dependence of air temperature, soil,

and vegetation on elevation and the strong lateral water-heat fluxes that occur at different scales from the latitudinal effect (Noetzli et al., 2017). At a global scale, climate aridity affects the distribution of permafrost primarily by climatic continentality. This reflects the vulnerability of permafrost dependence on the annual mean temperature range and variation in the net effect of precipitation and evapotranspiration. Here, we investigate the distribution of the permafrost region and permafrost area along latitude, elevation, and aridity transects in the NH (Figure 8).

The permafrost distribution depends strongly on latitude. In the NH, permafrost occurs from 28 °N in the Himalayas to north of 75 °N in Greenland and the Canadian Arctic Archipelago, and more than 90% is distributed in the regions north of 46 °N, predominantly between 52 °and 74 °N (Figure 8a). Approximately 8% of the existing permafrost regions are distributed in the regions south of 45 °N, mainly in the high-elevation regions of Asia, such as the Qinghai-Tibet Plateau between 28 °and 40 °N. With increasing latitude, the fraction of permafrost regions in the currently exposed land surface increases until approximately

70 °N, where the fraction begins to decrease with the sharp increase in Arctic oceans, glaciers and ice sheets. The profile of the permafrost area is generally consistent with that of the permafrost region. The decrease in the differences in the distributive trends and patterns of the permafrost region and permafrost area with rising northern latitude indicates the latitudinal dependence of permafrost areal continuity (PEC). The PEC is greater than 90% north of approximately 66 °N, i.e., terrestrial permafrost becomes continuous north of 66 °N.

The permafrost distribution also clearly depends on elevation, especially in the mountainous and high-plateau regions at middle and low latitudes. Approximately 80% of permafrost occurs below 1,000 m asl, and less than 10% of permafrost occurs above 3,000 m asl. In the areas below 3,000 m asl, the area of permafrost decreases with increasing elevation, but its fraction of the land surface is approximately stable. In the areas above 3,000 m asl, the area of permafrost and its fraction of the exposed land surface both increase with rising elevation (Figure 8b). This may indicate that latitude is the main controller of permafrost 390 distribution below 3,000 m asl, while elevation is a more significant controller above 3,000 m asl.

Regarding the profile of permafrost distribution along northern longitude (Figure 8c), our study results are generally consistent with those of existing studies (Zhang et al., 2008) but provide more spatial detail.

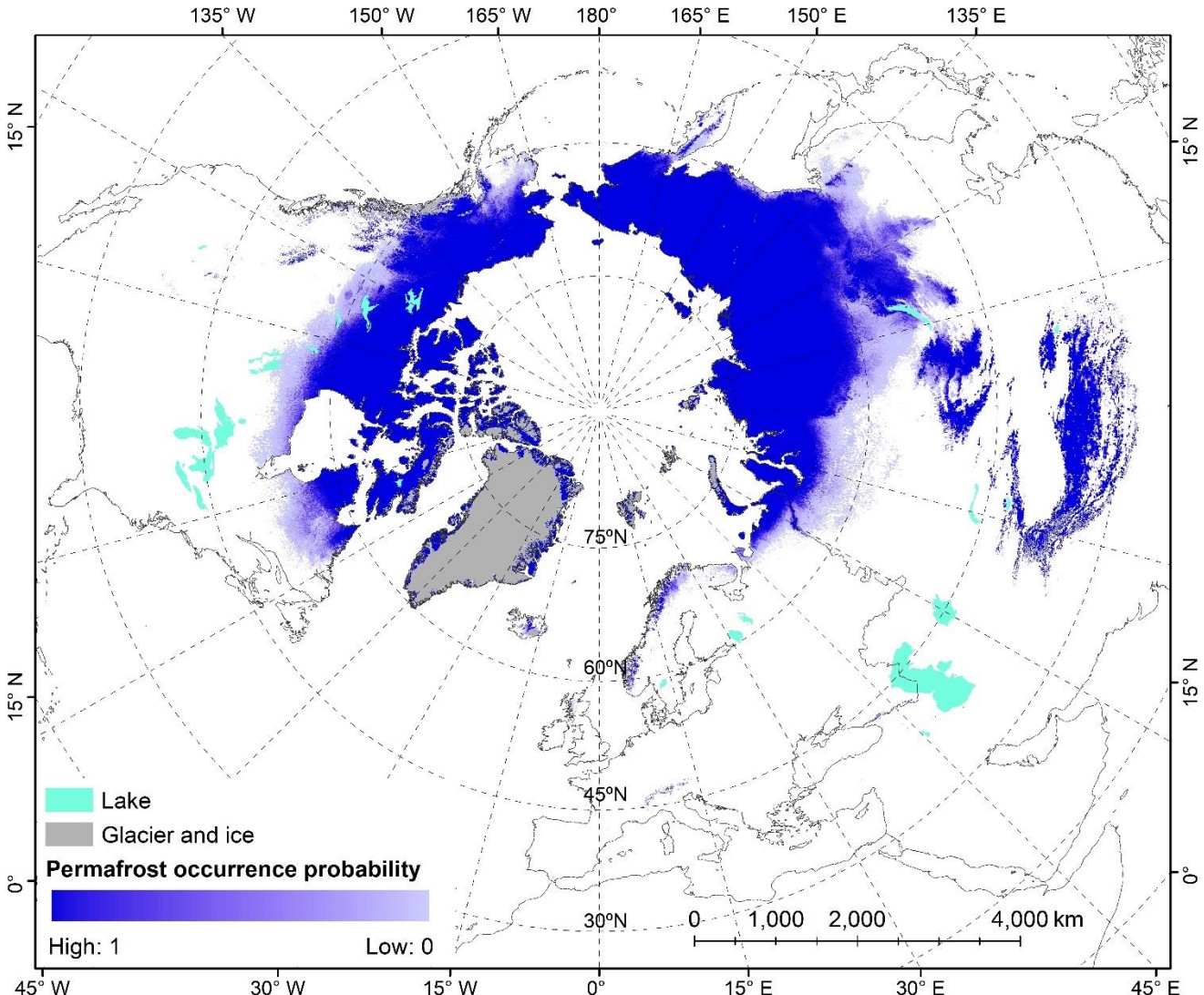

Figure 6: The present-day probability of permafrost occurrence in the Northern Hemisphere for the period of 2000-2016. (The
probability is defined by the fraction of predicted MAGT≤0 °C based on the multimodel ensemble results with 1,000 runs.)

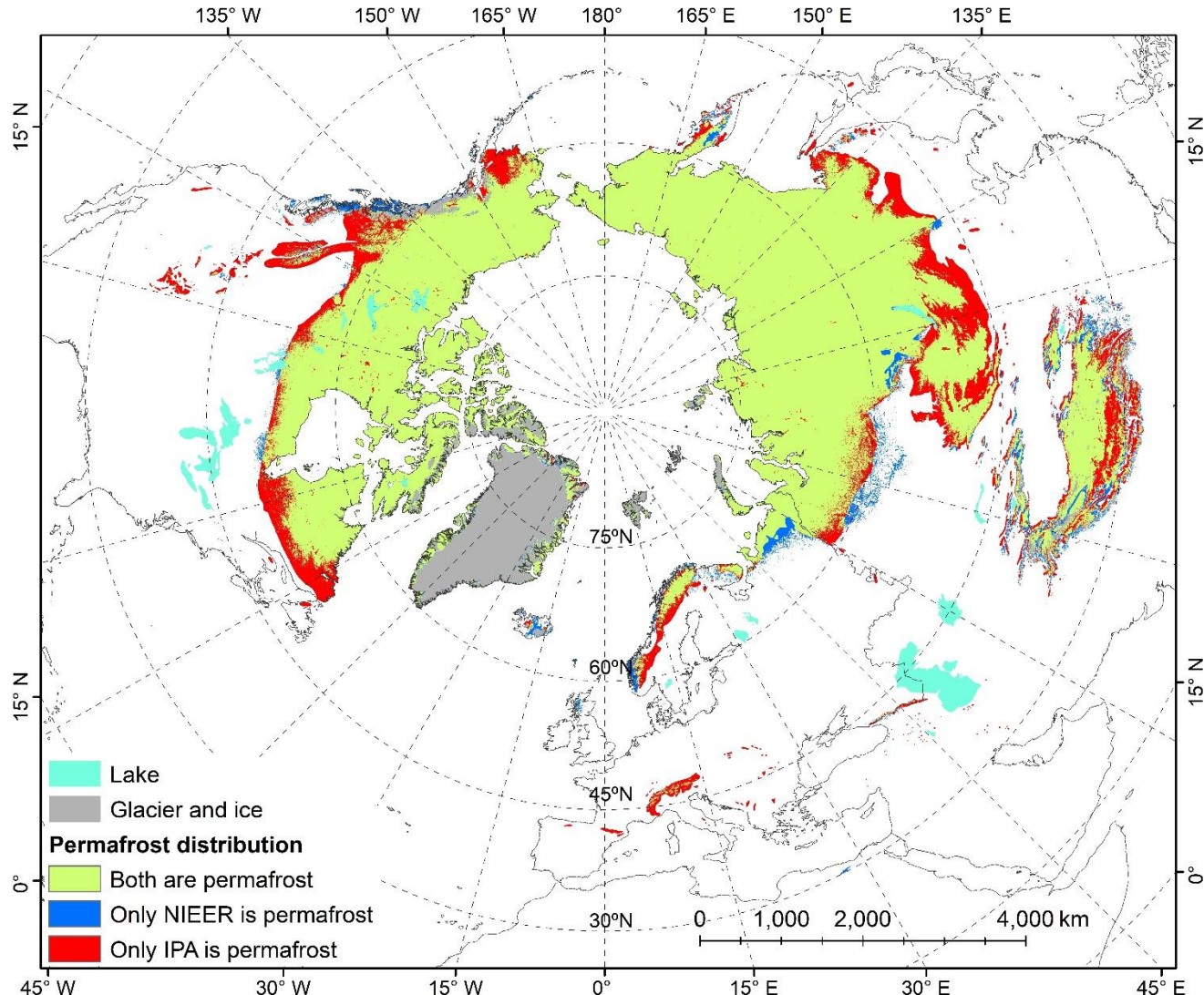

**Figure 7: The spatial difference of permafrost region proposed in this study (defined as the regions where the probability of permafrost occurrence is greater than 0) with Circum-Arctic permafrost map from International Permafrost Association (Brown et al., 1997).**

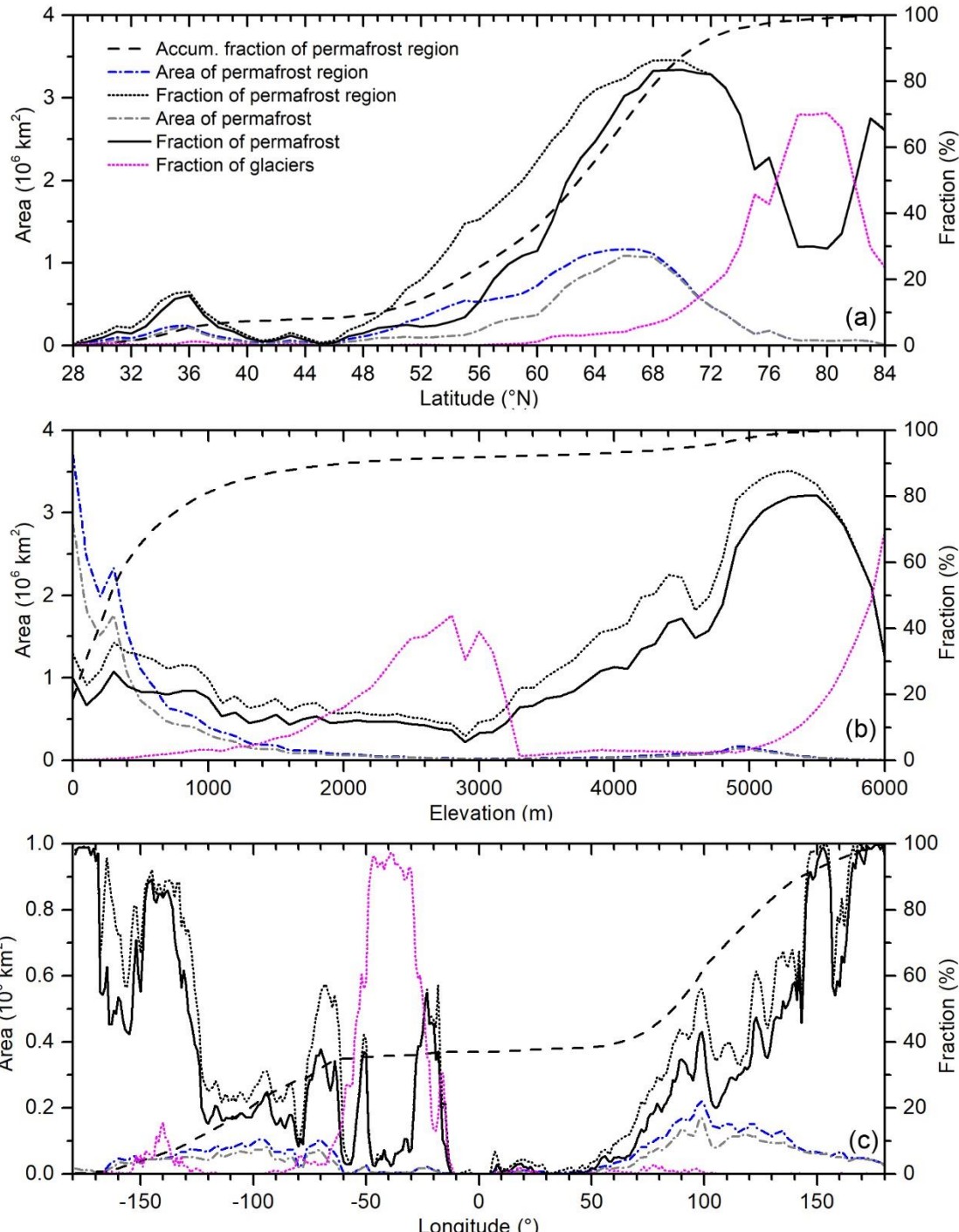

**Figure 8: Distributions of the permafrost region (permafrost probability>0) and permafrost area (MAGT≤0 °C) along northern latitude (a), elevation (b), and longitude (c) transects in the Northern Hemisphere (blue short dash-dot line: area of permafrost region, grey short dash-dot line: area of permafrost area, black dotted line: fraction of permafrost region, black dashed line:**

**accumulation fraction of permafrost region, black solid line: fraction of permafrost area, magenta dotted line: fraction of glacier area).**

### 3.3 Permafrost hydrothermal conditions

The hydrothermal conditions of permafrost differ remarkably over the NH, from the warm-arid type dominating on the Qinghai-Tibet Plateau to the cold-humid type dominating in the High Arctic. Figure 9 shows the distribution of permafrost

hydrothermal conditions in the NH. In the Eurasian Arctic, permafrost occurs mainly as the cold-humid type in Western Siberia and as the cold-semiarid/subhumid type in Central and Eastern Siberia. While cold-semiarid/subhumid permafrost prevails in the western Canadian Arctic, the eastern Canadian Arctic contains cold-humid permafrost. The cold-humid type is dominant in Greenland. In the Alaskan Arctic, there is cold-semiarid/subhumid permafrost. In the permafrost regions at middle and low latitudes, such as the Qinghai-Tibet and Mongolian plateaus, permafrost occurs mainly as the warm-arid type. Warm-humid

permafrost mainly occurs to the south of the continuous permafrost zone in Western Siberia and eastern Canada.

In general, regarding moisture conditions, permafrost in the NH is dominated by the humid type, which accounts for approximately 51.56% ($10.22 \times 10^6$ km$^2$) of the permafrost region. The areas of semiarid/subhumid and arid permafrost regions account for 45.07% ($8.93 \times 10^6$ km$^2$) and 3.37% ($0.67 \times 10^6$ km$^2$) of the region, respectively (Table 3). Regarding thermal conditions, the areas of cold, cool, and warm permafrost regions are approximately $7.49 \times 10^6$ (37.80%), $2.84 \times 10^6$ (14.30%),

and $9.49 \times 10^6$ km$^2$ (47.90%), respectively. The hydrothermal conditions are closely related to the ecosystem and climate. Thus, it is helpful to understand the environmental changes in permafrost regions and their interaction with permafrost degradation.

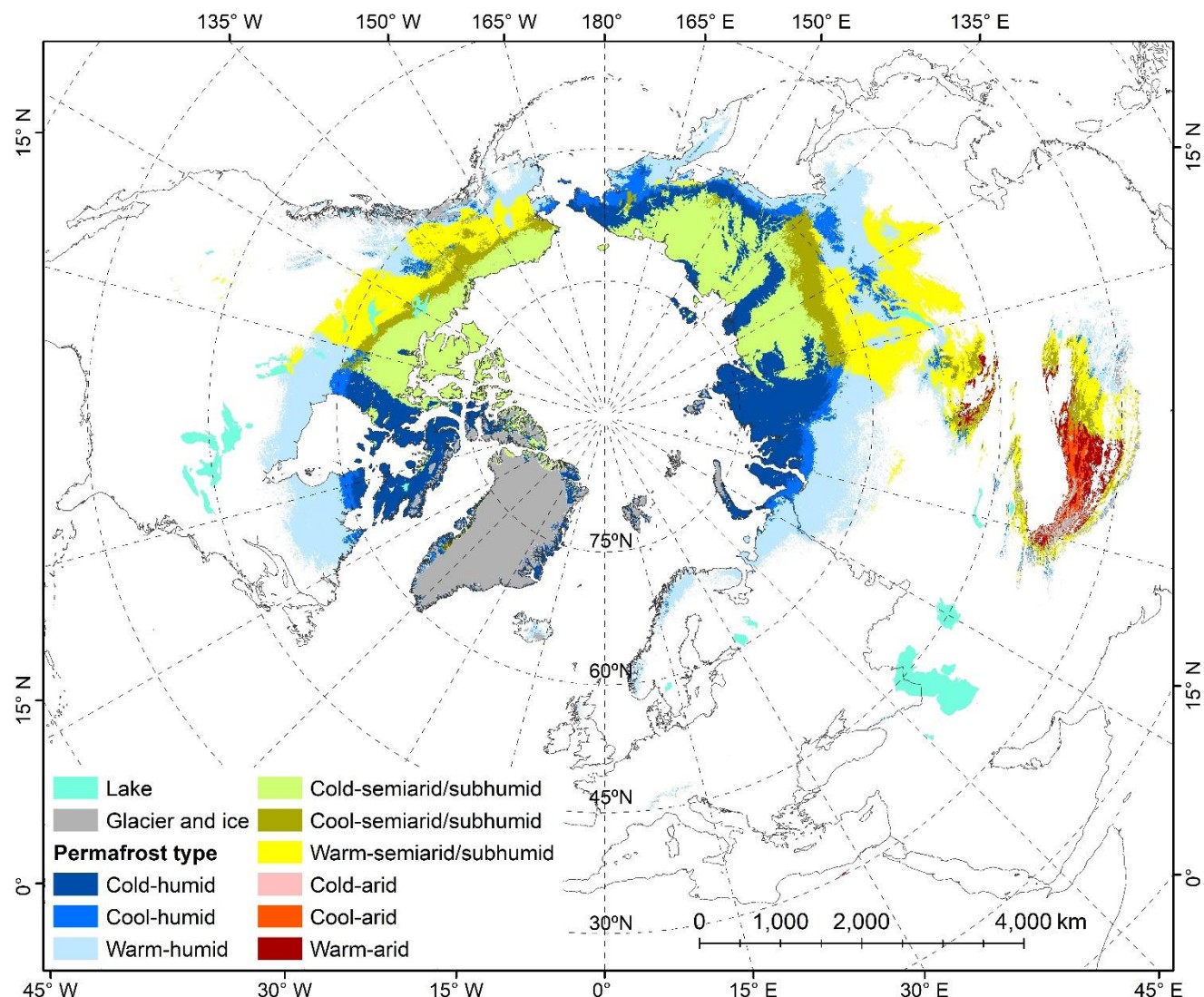

**Figure 9: Hydrothermal condition-based permafrost zonation in the Northern Hemisphere for the period of 2000-2016.**

**Table 3: The zonal areas of different permafrost hydrothermal conditions ($10^6$ km$^2$) over the currently exposed land surface in the Northern Hemisphere.**

| Thermal state of permafrost | Humid (AI>0.65) | Semiarid/subhumid (0.65≥AI>0.20) | Arid (AI≤0.20) | Total |
|---|---|---|---|---|
| Cold (MAGT≤−3 ℃) | 3.56 | 3.89 | 0.04 | 7.49 |
| Cool (−3.0 ℃<MAGT≤−1.5 ℃) | 1.20 | 1.44 | 0.20 | 2.84 |
| Warm (MAGT>−1.5 ℃) | 5.46 | 3.60 | 0.43 | 9.49 |
| Total | 10.22 | 8.93 | 0.67 | 19.820 |

**4 Data availability**

The NIEER datasets generated by this study (Table 4) are publicly available and can be downloaded at the National Tibetan Plateau Data Center (TPDC) (http://data.tpdc.ac.cn/en) via https://data.tpdc.ac.cn/en/data/5093d9ff-a5fc-4f10-a53f-c01e7b781368 or https://doi.org/10.11888/Geocry.tpdc.271190 (Ran et al., 2021b). The MAGT, ALT, and permafrost
probability in GeoTiff format and the permafrost hydrothermal zonation map in ESRI shapefile format can be used with GIS software.

**Table 4: The list of permafrost datasets for the Northern Hemisphere produced in this study (MAGT: mean annual ground temperature, ALT: active layer thickness, GAM: generalized additive model, SVR: support vector regression, RF: random forest, and XGB: eXtreme gradient boosting).**

| Name | Unit | Description |
|---|---|---|
| MAGT | ℃ | An ensemble average of four statistical techniques (GAM, SVR, RF, and XGB) with 1,000 runs |
| ALT | cm | An ensemble average of four statistical learning techniques (GAM, SVR, RF, and XGB) with 1,000 runs |
| Permafrost probability | - | The fraction of MAGT≤0 ℃ based on the 1,000 multimodel ensemble of MAGT predictions |
| Permafrost zonation (hydrothermal conditions) | % | The hydrothermal condition-based permafrost zonation after the processing described in Section 2.5 |

**5 Conclusions**

This study produced MAGT at or near the DZAA and ALT datasets with 1-km resolution for the period of 2000–2016 in the NH (named the NIEER map). The datasets integrate unprecedentedly large amounts of ground/field measurement data (1,002 boreholes for MAGT and 452 sites for ALT) and multisource spatial data, including remotely sensed FDD and TDD, LAI, SCD, precipitation, solar radiation, soil organic content, soil bulk density, and coarse fragment content, using a multiple
statistical/machine learning model with 1,000 runs. The cross-validation shows that the accuracy of the MAGT (bias=0.02±0.16 ℃, RMSE=1.32±0.13 ℃) and ALT (bias=2.71±16.46 cm, RMSE=86.93±19.61 cm) data is likely greater than that in previous studies. The NIEER map suggests that the areal extent of permafrost (MAGT≤0 °C) in the NH, excluding glaciers and lakes, is approximately 14.77 (13.60–18.97) $\times 10^6$ km², which is generally consistent with recent reports. Furthermore, the area of permafrost regions (permafrost probability>0) (approximately 19.82 $\times 10^6$ km²) in the NIEER map is
less than that estimated in the IPA circum-Arctic permafrost map. This difference likely reflects the difference in cartography technology between the two eras. Based on the new permafrost classification system proposed in this study, a hydrothermal condition-based permafrost zonation map was produced by combining the permafrost probability map with MAGT and the aridity index to describe the hydrothermal characteristics of permafrost in the NH. The results show that the areal fractions of humid, semiarid/subhumid, and arid permafrost regions are 51.56%, 45.07%, and 3.37%, respectively. The areal fractions of

cold (≤-3.0 ℃), cool (-3.0 ℃ to -1.5 ℃), and warm (>-1.5 ℃) permafrost regions are 37.80%, 14.30%, and 47.90%, respectively. Based on high-quality datasets, this study has provided a more comprehensive and accurate understanding of the permafrost distribution in the NH through analyses of changes in the MAGT, ALT, and permafrost area/region along latitude, elevation, and aridity index transects. The model-predicted MAGT and ALT as well as the corresponding permafrost probability map and hydrothermal zonation map are potentially valuable for studies of cold regions and the Arctic in a variety

of fields, such as climatology, hydrology, ecology, agriculture, public health, and engineering planning in the NH. Additionally, as baselines, these datasets are also important for predicting and rebuilding changes in permafrost features over the NH in the future and in the past.

## Appendix A:   List of mean annual ground temperature (MAGT) ground measurement data sources

**Table A1.** Data sources used to compile mean annual ground temperature (MAGT) datasets for the period 2000–2016. The number of observations, minimum, mean and maximum depths of MAGT and proportions of MAGT measured at or near the depth of zero annual amplitude (DZAA) are provided for each sources.

| Data source | Reference | MAGT observations | | | | | |
|---|---|---|---|---|---|---|---|
| | | Number | Depth (m) | | | At or near DZAA (%) | |
| | | | min | mean | max | yes | no |
| GTN-P Database | Biskaborn et al. (2015) | 461 | 2 | 12.3 | 37 | 78 | 22 |
| National Snow & Ice Data Center, Boulder, Colorado, USA | Paetzhold (2003) | 2 | 2.7 | 10.4 | 18 | 50 | 50 |
| Roshydromet | Sherstiukov (2012) | 91 | 3.2 | 3.2 | 3.2 | 0 | 100 |
| Geological Survey of Canada | Smith et al. (2013) | 50 | 5 | 20.1 | 36 | 92 | 8 |
| | Crow et al. (2015) | 46 | 13 | 16.2 | 40.1 | 100 | 0 |
| | Smith and Ednie, (2015) | 7 | 1.9 | 5.8 | 9.9 | 57 | 43 |
| | Ednie et al. (2012) | 1 | 15 | 15 | 15 | 100 | 0 |
| | Wolfe et al. (2010) | 1 | 15 | 15 | 15 | 100 | 0 |
| National Geothermal Data System, U.S. Department of Energy | Blackett (2013) | 18 | 15 | 16.1 | 20 | 100 | 0 |
| | Maine Geological Survey (2014) | 19 | 11.9 | 16.2 | 30.8 | 100 | 0 |
| | Kelley (2011) | 11 | 15 | 15 | 15 | 100 | 0 |
| | Virginia Division of Geology and Mineral Resources (2012) | 1 | 15.2 | 15.2 | 15.2 | 100 | 0 |
| | Curran et al. (2013) | 5 | 15.2 | 15.3 | 15.3 | 100 | 0 |
| | Czajkowski (2012) | 8 | 14.9 | 15 | 15 | 100 | 0 |
| | Virginia Division of Geology and Mineral Resources (2012) | 6 | 15 | 15 | 15 | 100 | 0 |
| | University of North Dakota (2014) | 4 | 15 | 15 | 15 | 100 | 0 |
| | Gosnold (2013) | 1 | 15 | 15 | 15 | 100 | 0 |
| | Niewendorp (2012) | 1 | 25 | 25 | 25 | 100 | 0 |
| | Harrison III (2012) | 1 | 15.2 | 15.2 | 15.2 | 100 | 0 |
| | Nevada Bureau of Mines and Geology (2014) | 1 | 15.2 | 15.2 | 15.2 | 100 | 0 |
| National Oceanic and Atmospheric Administration, U.S. Department of Commerce | Huang et al. (2000) | 13 | 16 | 27.8 | 44 | 100 | |
| Finnish Meteorological Institute | | 9 | 2 | 2.3 | 4 | 0 | 100 |
| NSF Arctic Data Center | NSF Arctic Data Center (2014) | 3 | 3 | 3 | 3 | 0 | 100 |
| Nordicana D, Centre for Northern Studies | Allard et al. (2015) | 3 | 9.5 | 13.2 | 15 | 67 | 33 |
| The Geophysical Institute Permafrost Laboratory (GIPL), University of Alaska, Fairbanks | GIPL (2010) | 2 | 11 | 12.5 | 14 | 100 | 0 |
| Publications | Ødegård et al. (2008) | 2 | 8.5 | 8.5 | 8.5 | 100 | 0 |
| | Streletskiy et al. (2015) | 2 | 10 | 10 | 10 | 100 | 0 |
| | Peter (2015) | 2 | 3.8 | 3.9 | 4 | 0 | 100 |

| | | | | | | |
|---|---|---|---|---|---|---|
| | Cao et al. (2017) | 12 | 11.5 | 15.4 | 16 | 100 | 0 |

| | | | | | | |
|---|---|---|---|---|---|---|
| | Cao et al. (2017) | 12 | 11.5 | 15.4 | 16 | 100 | 0 |
| | Chang (2011) | 15 | 12 | 16.7 | 20 | 100 | 0 |
| | Li et al. (2011) | 15 | 8 | 12.7 | 15 | 100 | 0 |
| | Li et al. (2016) | 35 | 15 | 17.1 | 20 | 100 | 0 |
| | Li et al. (2019) | 3 | 16 | 16 | 16 | 100 | 0 |
| | Luo (2012) | 16 | 7 | 15.8 | 42 | 100 | 0 |
| | Luo et al. (2018) | 3 | 15 | 15 | 15 | 100 | 0 |
| | Liu et al. (2015) | 16 | 10 | 10 | 10 | 100 | 0 |
| | Qin et al. (2017) | 4 | 10 | 12 | 15 | 100 | 0 |
| | Sun et al. (2013) | 1 | 18 | 18 | 18 | 100 | 0 |
| | Sheng et al. (2010) | 10 | 8 | 12.2 | 15 | 100 | 0 |
| | Yu et al. (2013) | 5 | 15 | 15 | 15 | 100 | 0 |
| | Wani et al. (2020) | 3 | 10 | 10 | 10 | 100 | 0 |
| | Wang et al. (2013) | 2 | 16 | 18 | 20 | 100 | 0 |
| | Wu et al. (2007) | 6 | 10 | 10 | 10 | 100 | 0 |
| | Wu et al. (2015) | 10 | 12 | 14.3 | 15 | 100 | 0 |
| | Yu et al. (2008) | 37 | 18 | 18 | 18 | 100 | 0 |
| | Zhao et al. (2021) | 26 | 10 | 10 | 10 | 100 | 0 |
| | Zhang et al. (2011) | 11 | 10 | 10 | 10 | 100 | 0 |
| Geological Survey of Norway (NGU) | NGU (2016) | 1 | 15 | 15 | 15 | 100 | 0 |
| TOTAL (in permafrost region) | | 713 | 1.9 | 12.84 | 42 | 84 | 16 |
| TOTAL (all sites) | | 1002 | 1.9 | 13.62 | 44 | 79 | 21 |

**Author contributions:**

Y.R., X.L., and G.C. designed the study. Y.R. prepared the datasets, wrote the manuscript, plotted the figures and performed the analysis. Y.R. designed the methodology, and J.C. implemented the statistical learning. J.A., O.K., J.H., M.L., Q.Y., and C.X. contributed parts of the field data. J.O. provided the TDD and FDD data. X.L., J.A., O.K., J.H., M.L., and H.J. improved the writing and structure of the paper.

**Competing interests:**

The authors declare that they have no conflicts of interest.

**Acknowledgements:**

This study was jointly supported by the Strategic Priority Research Program of the Chinese Academy of Sciences (Grant No. XDA19070204) and the National Natural Science Foundation of China (Grant No. 42071421). O.K. and J.H. acknowledge support from the Academy of Finland (Grant No. 315519). FDD and TDD data were generated by the European Space Agency GlobPermafrsot project (Grant No. 4000116196/15/I-NB).

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
