# Peer review of "New high-resolution estimates of the permafrost thermal state and hydrothermal conditions over the Northern Hemisphere"

_Earth System Science Data, 2021_

## Editor Comment (EC1)

Dear Joshua Ran and co-authors,

I have received a new comment to your manuscript from Dr. Robert Way which is highly relevant for the data product you are describing and needs to be addressed during the review (see the supplement)

Kind Regards

Kirsten Elger

Original comment of Dr. Robert Way (Queen's University, Canada; robert.way@queensu.ca)

To the authors of Ran et al (in review - http://doi.org/10.5194/essd-2021-83). This is an interesting approach where the authors are combining the Obu et al (2019) FDD/TDD products with advanced machine learning techniques and GTNP datasets to estimate permafrost characteristics and distribution across the Northern Hemisphere. The approach is interesting and has some utility so for that the authors certainly deserve praise. However, I am left with the same thoughts that I have had about many of these empirical-statistical modelling approaches that have accelerated in the permafrost community.

My concern is largely that the increasing number of statistical variables being used in these products does not make up for the lack of quality of the input products. Crucially in the case of two of the most important variables: snow thickness and surficial materials type. In the former case, the authors use snow cover duration as a supposed proxy which many on the author team know is not a suitable replacement for snow thickness. In the latter case, the authors use the Soilgrids250 product which is quite similar to the author's approach methodologically and is a derivative of many of the same input parameters. Critically, the latter product does not include any representation of the glaciomarine limits or of regional glacial depositional history making it unsuitable for specific uses such as permafrost risk mapping in coastal areas. Finally, the large gaps in the regions represented in the statistical modelling mean that the regional variations in the importance of certain other factors is likely to be missed. Alterations to the thermal impact of temperature range was brought up by Dan Riseborough's thesis where lower temperature ranges at similar MAATs can produce colder MAGTs. The geostatistical approaches used in these types of papers would undoubtedly miss this 'process-based' explanation leading to systematic biases.

I can only speak to the regions that I have worked in (Yukon / Québec / Labrador) but I do not find this distribution map (nor many predecessors using similar empirical-statistical approaches) to adequately represent permafrost distribution in Subarctic and coastal northern Canada. This is particularly the case in Québec and Labrador where the lack of inclusion permafrost input data or any metrics of snow thickness / coastal geomorphology leads to a permafrost distribution map that is at odds with our regional understanding. There have been a number of publications which suffer from similar challenges (e.g. Aalto et al 2018; Hjort et al 2018; Karjalainen et al. 2019; Hugelius et al 2020; Olefeldt et al 2016) owing to the heavy reliance

on MAAT/FDD/TDD as a proxy for permafrost distribution without incorporation of the mediating effects of snow thickness. The importance of coastal effects (mostly via wind scouring, temperature amplitude and snow density changes) and coastal geomorphology (marine limit) is also a challenge in the Labrador region as we find permafrost far more likely to occur in the coastal areas versus the interior - opposite to what is seen in this distribution map and those of a variety of other products.

Representing snow thickness and surficial materials are significant challenges that will take time for various products to be generated but using more variables with more advanced statistical techniques is not necessarily a substitute for better quality datasets needed for this type of mapping initiative. In my specific region (which the authors included no data from) these issues lead to a completely wrong representation of permafrost distribution. The newly developed Ground Ice Map of Canada shows a pathway forward for including process-based knowledge in large-scale map generation and I would strongly urge the author team to consider lessons from that product.

Normally, I would not comment on these efforts but admittedly these products are now being used by other studies (e.g. Hjort et al 2018) for infrastructure hazard assessments and it is leading to a misinterpretation of the regions at risk of permafrost thaw. Producing maps of permafrost distribution or infrastructure risk to thaw that are at odds with our detailed understanding of the drivers of permafrost distribution **does have ethical implications** and I would ask the author team to reflect on whether the products being produced should include regions for which these errors are likely to be exasperated. As a co-author of Obu et al (2019), I do also have to acknowledge that I think our paper is also being regularly misused in a similar manner.

As two side notes: (1) Using DZAA data for calibration may mean that thinner permafrost bodies are missed leading to an underestimation of the distribution of permafrost in marginal environments. We observe this at all of our sites in southeastern Labrador (Way et al. 2018). (2) There should be a rationale provided for the use of FDD/TDD from Obu et al (2019) compared to alternatives from gridded climate products from meteorological stations or downscaled reanalysis. I would challenge the authors to evaluate whether these MODIS-based datasets are an improvement in representing air temperatures compared to meteorological products (most studies in North America do not support MODIS inclusion being unbiased).

I will end by stating that the criticism of this paper is not meant to be targeted to these authors as I think this is more of an expanded set of thoughts that extends to this overall methodology. Can we produce (overfit) estimates of various permafrost parameters using random assortments of inputs of varying quality? Sure, we can. But the question is should we? And if we do produce these, what is the value of doing so and what are the ethical implications of not making clear to users that these products are not suitable for regional scale analyses?

Dr. Robert Way

Department of Geography and Planning

Queen's University, Canada (robert.way@queensu.ca)

References

Obu, J., Westermann, S., Bartsch, A., Berdnikov, N., Christiansen, H.H., Dashtseren, A., Delaloye, R., Elberling, B., Etzelmüller, B., Kholodov, A., Khomutov, A., Kääb, A., Leibman, M.O., Lewkowicz, A.G., Panda, S.K., Romanovsky, V., Way, R.G., Westergaard-Nielsen, A., Wu, T., Yamkhin, J., and Zou, D. 2019. Northern Hemisphere permafrost map based on TTOP modelling for 2000–2016 at 1 km2 scale. Earth-Science Reviews, 193: 299–316. doi:10.1016/j.earscirev.2019.04.023.

Hugelius, G., Loisel, J., Chadburn, S., Jackson, R.B., Jones, M., MacDonald, G., Marushchak, M., Olefeldt, D., Packalen, M., Siewert, M.B., Treat, C., Turetsky, M., Voigt, C., and Yu, Z. 2020. Large stocks of peatland carbon and nitrogen are vulnerable to permafrost thaw. Proceedings of the National Academy of Sciences, 117: 20438–20446. doi:10.1073/pnas.1916387117.

Hjort, J., Karjalainen, O., Aalto, J., Westermann, S., Romanovsky, V.E., Nelson, F.E., Etzelmüller, B., and Luoto, M. 2018. Degrading permafrost puts Arctic infrastructure at risk by mid-century. Nature Communications, 9. doi:10.1038/s41467-018-07557-4.

Aalto, J., Karjalainen, O., Hjort, J., and Luoto, M. 2018. Statistical Forecasting of Current and Future Circum-Arctic Ground Temperatures and Active Layer Thickness. Geophysical Research Letters,. doi:10.1029/2018GL078007.

Olefeldt, D., Goswami, S., Grosse, G., Hayes, D., Hugelius, G., Kuhry, P., McGuire, A.D., Romanovsky, V.E., Sannel, A.B.K., Schuur, E.A.G., and Turetsky, M.R. 2016. Circumpolar distribution and carbon storage of thermokarst landscapes. Nature Communications, 7: 13043. doi:10.1038/ncomms13043.

Way, R.G., Lewkowicz, A.G., and Zhang, Y. 2018. Characteristics and fate of isolated permafrost patches in coastal Labrador, Canada. The Cryosphere, 12: 2667–2688. doi:10.5194/tc-12-2667-2018.

---

## Author Comment (AC1)

<h1 style="text-align:center">Responses to Reviewer's Comments and Suggestions</h1>
(The answers are shown in blue)

**Referee comment:**

The authors of the manuscript have done a lot of work on mapping the thermal state of the permafrost in the Northern Hemisphere: the average annual soil temperature (MAGT) at a depth of zero annual amplitude (ZAA) and the thickness of the active layer (ALT), zoning of permafrost based on hydrothermal conditions with a resolution of 1 km per period 2000-2016 The results obtained undoubtedly enrich the previously obtained data of previous researchers and have been successfully mapped on a global scale (the entire northern hemisphere), they can also be mapped at the local (regional) level, as the most demanded in practice.

**Technical Notes (Corrections):**

- In fig. 2 contours of the "lake" are missing.? They are present in the following figures. This error should be eliminated;

Response: Thank you. The "lake" will be added in the revised manuscript.

- In Fig. 3, the MAGT (oC) color selection is unsuccessful (-15 - -14; -14 - -13 and -2 - -1; -1 - 0). They almost do not differ in color. Also, the color of the "lake" repeats the color "-10…-9; -9 - -8". You should choose a different color for the "lake";

Response: We will adjust the color scheme to improve the discrimination.

- In fig. 4 remark is similar as in fig. 3. Choose a different shade for the color of the "lake".

Response: We will adjust the color of the "lake".

In my opinion, the gradations of average annual temperatures should be chosen according to generally accepted classifications of types of seasonal freezing and thawing of rocks: transitional (-1 - 0), semi-transitional (-2 - -1), long-term stable (-5 - -2), stable (-10 - 5), arctic (-15 - -10) and polar (-20 - -15). I do not insist on following my advice, leaving the choice to the authors of the manuscript.

Response: Thank you for your suggestion. This legend is mainly to show the distribution of MAGT in detail, so a detailed classification interval is used here. We will optimize the color configuration to improve its discrimination.

---

## Author Comment (AC2)

Responses to Reviewer's Comments and Suggestions
(The answers are shown in blue)

**Review comments on ESSD-2021-83**

The authors probably compiled the most up-to-date dataset for permafrost and active layer thickness available worldwide. This first hand data and information would definitely make the mapping of permafrost and active layer thickness more accurate and reliable. The authors proposed new principal approach of permafrost mapping by using both mean annual ground temperature and aridity, this is new and very creative. The newly created permafrost map and dataset of active layer thickness would be an extremely valuable for cold regions and Arctic studies in variety of fields. I recommend the manuscript be accepted for publication in ESSD with some minor revisions:

1. L185: provide the "SoilGrids250" website or data center.

Response: The website of "SoilGrids250" will be provided in the revised manuscript.

2. L327: "….and aridity transects in the NH, as shown in Figure 7." Should be "….and aridity transects in the NH (Fig. 7).

Response: Revised as suggested.

3. As the authors mentioned several times, the IPA map was a milestone for NH permafrost mapping. I would suggest that the authors conduct a detailed comparison to see the spatial difference between two maps. It seems the IPA map is a little over estimate area of permafrost regions, where? The authors may conduct a spatial difference figure between the two maps.

Response: According to your helpful suggestion, a more details comparison include a spatial difference figure will be added in revised manuscript.

4. The authors may also explain why the two maps are different. First, the IPA map was a mechanical compilation of national maps, each nation had their own mapping standard, it therefore brings a lot of errors and uncertainties. Second, the new map used MAGT<=0 ℃ as the boundary. In a real world, this is a very restrictive requirement. Due to the effect of thermal offset within the active layer, there may exist permafrost between the depth of ZAA and the depth of seasonal maximum thaw (ALT). There is may be other reasons, the authors do not need to do any new work but discuss potential issues in the text.

Response: We will added some discussion for the different between IPA map and our new map.

5. The IPA map statistics may not exclude lakes in permafrost regions. However, there are numerous lakes in permafrost regions, the authors should provide information on what size of lakes were excluded from their new map although Fig. 6 has shown the excluded lakes.

Response: This will be added in Section 3.2.

6. "Climate aridity" is a more reflecting the distance from oceans rather than longitudinal. The authors may just consider changes in words in the text.

Response: Thank you. This have been corrected.

7. I just wonder if the models used by the authors can output TTOP?

Response: The output of the models is dependent on training data, MAGT is used here, so TTOP cannot be output at present.

8. Bin Cao et al. also did the similar work over Qinghai-Tibetan Plateau. I recommend that the authors should include these work in their review.

Response: This reference will been added.

9. 3: Be clear the figure shows the MAGT average over period of 2000-2016, i.e. "the average MAGT" or "mean MAGT".

Response: Yes. We will clear this.

10. 4: save as the above. "the average active layer thickness"

Response: This have been corrected.

11. 5: explain in details on what on the figure, such as what is black solid line? What is the dashed line? What is the shaded area and overlapped shaded area? Most readers may understand but it needs to be clear in caption.

Response: More details explain will be added to clear this.

12. 6: The authors may need to distinguish their probability with the previous studies such as Gruber et al. (2012) and Cao (2018??).

Response: Yes, these are two different definitions. We will clear this.

13. 7: The authors need to describe each line in the Figure in the Caption. Just from legend alone, it is hard to know what is what.

Response: More details explain will be added to clear this.

14. All major results in the Abstract are not in Conclusions. These major results should be in more detailed than in the Abstract. Conclusions need to be expanded. Very often, potential readers read the title first, then the Abstract, then Conclusions, Figures with detailed captions, then the whole text depending on their interests.

Response: According to your very helpful comments, the conclusion have been expanded and other text and caption of figures are also improved.

15. Again, the authors' permafrost map has its probability>0, this need to be clarified when comparing with the IPA map. The authors should also include statistics of areas with MAGT<0.0C, their probability map, and the IPA map. It will be interesting to their difference and why?

Response: Thank you. We will clear this.

---

## Author Response (AR1)

**Responses to Reviewer's Comments and Suggestions**
**(The answers are shown in blue)**

**Referee 1 comments**

The authors of the manuscript have done a lot of work on mapping the thermal state of the permafrost in the Northern Hemisphere: the average annual soil temperature (MAGT) at a depth of zero annual amplitude (ZAA) and the thickness of the active layer (ALT), zoning of permafrost based on hydrothermal conditions with a resolution of 1 km per period 2000-2016 The results obtained undoubtedly enrich the previously obtained data of previous researchers and have been successfully mapped on a global scale (the entire northern hemisphere), they can also be mapped at the local (regional) level, as the most demanded in practice.

**Technical Notes (Corrections):**

- In fig. 2 contours of the "lake" are missing.? They are present in the following figures. This error should be eliminated;

Response: Thank you. The "lake" has been added in the revised manuscript.

- In Fig. 3, the MAGT (oC) color selection is unsuccessful (-15 - -14; -14 - -13 and -2 - -1; -1 - 0). They almost do not differ in color. Also, the color of the "lake" repeats the color "-10…-9; -9 - -8". You should choose a different color for the "lake";

Response: We have adjusted the colour scheme to improve its visibility.

- In fig. 4 remark is similar as in fig. 3. Choose a different shade for the color of the "lake".

Response: We have adjusted the colour of the "lake".

In my opinion, the gradations of average annual temperatures should be chosen according to generally accepted classifications of types of seasonal freezing and thawing of rocks: transitional (-1 - 0), semi-transitional (-2 - -1), long-term stable (-5 - -2), stable (-10 - 5), arctic (-15 - -10) and polar (-20 - -15). I do not insist on following my advice, leaving the choice to the authors of the manuscript.

Response: Thank you for your suggestion. This legend mainly shows the distribution of MAGT in detail, so a detailed classification interval is used here. We have optimized the colour configuration to improve its discrimination.

**Referee 2 comments**

The authors probably compiled the most up-to-date dataset for permafrost and active layer thickness available worldwide. This first hand data and information would definitely make the mapping of permafrost and active layer thickness more accurate and reliable. The authors proposed new principal approach of permafrost mapping by using both mean annual ground temperature and aridity, this is new and very creative. The newly created permafrost map and dataset of active layer thickness would be an extremely valuable for cold regions and Arctic studies in variety of fields. I recommend the manuscript be accepted for publication in ESSD with some minor revisions:

1.  L185: provide the "SoilGrids250" website or data center.

Response: The website "SoilGrids250" has been provided in the revised manuscript.

2.  L327: "….and aridity transects in the NH, as shown in Figure 7." Should be "….and aridity transects in the NH (Fig. 7).

Response: This has been revised as suggested.

3.  As the authors mentioned several times, the IPA map was a milestone for NH permafrost mapping. I would suggest that the authors conduct a detailed comparison to see the spatial difference between two maps. It seems the IPA map is a little over estimate area of permafrost regions, where? The authors may conduct a spatial difference figure between the two maps.

Response: According to your helpful suggestion, a more detailed comparison including a spatial difference figure has been added to the revised manuscript, as shown in Figure 7.

4.  The authors may also explain why the two maps are different. First, the IPA map was a mechanical compilation of national maps, each nation had their own mapping standard, it therefore brings a lot of errors and uncertainties. Second, the new map used MAGT<=0 °C as the boundary. In a real world, this is a very restrictive requirement. Due to the effect of thermal offset within the active layer, there may exist permafrost between the depth of ZAA and the depth of seasonal maximum thaw (ALT). There is may be other reasons, the authors do not need to do any new work but discuss potential issues in the text.

Response: Some discussion of the difference between the IPA map and our new map has been added in Section 3.2.

5.  The IPA map statistics may not exclude lakes in permafrost regions. However, there are numerous lakes in permafrost regions, the authors should provide

information on what size of lakes were excluded from their new map although Fig. 6 has shown the excluded lakes.

Response: This has been added in L320 in Section 3.2.

6. "Climate aridity" is a more reflecting the distance from oceans rather than longitudinal. The authors may just consider changes in words in the text.

Response: Thank you. This has been corrected.

7. I just wonder if the models used by the authors can output TTOP?

Response: The output of the models is dependent on training data. MAGT is used here, so TTOP cannot be output at present.

8. Bin Cao et al. also did the similar work over Qinghai-Tibetan Plateau. I recommend that the authors should include these work in their review.

Response: This reference has been added.

9. 3: Be clear the figure shows the MAGT average over period of 2000-2016, i.e. "the average MAGT" or "mean MAGT".

Response: Thank you for this suggestion. We have clarified this.

10. 4: save as the above. "the average active layer thickness"

Response: This has been corrected.

11. 5: explain in details on what on the figure, such as what is black solid line? What is the dashed line? What is the shaded area and overlapped shaded area? Most readers may understand but it needs to be clear in caption.

Response: More details have been added to clarify this.

12. 6: The authors may need to distinguish their probability with the previous studies such as Gruber et al. (2012) and Cao (2018??).

Response: Yes, these are two different definitions. We have clarified this.

13. 7: The authors need to describe each line in the Figure in the Caption. Just from legend alone, it is hard to know what is what.

Response: More details have been added to the figure caption to clarify this.

14. All major results in the Abstract are not in Conclusions. These major results should be in more detailed than in the Abstract. Conclusions need to be expanded. Very often, potential readers read the title first, then the Abstract, then Conclusions, Figures with detailed captions, then the whole text depending on their interests.

Response: According to your very helpful comments, the conclusion has been expanded, and other text and figure captions have also been improved.

15. Again, the authors' permafrost map has its probability>0, this need to be clarified when comparing with the IPA map. The authors should also include statistics of areas with MAGT<0.0C, their probability map, and the IPA map. It will be interesting to their difference and why?

Response: Thank you. We have clarified this in L332.

---

## Referee Report (RR1)

Review comments on ESSD-2021-83

Youhua Ran et al. present a relevant raster data collection of permafrost-related ground quantities, these are the GCOS Essential Climate Variables, ECVs: mean annual ground temperature MAGT at zero annual amplitude (ZAA), and active layer thickness (ALT) representative for a thermal state of permafrost for the time window from 2010 to 2016. In addition, the authors also derived permafrost probability and what is the novelty: aridity-index related permafrost regions.

Despite the high value of providing these mapped permafrost related quantities and the very interesting novel approach integrating the aridity, the manuscript still lacks clarity and accuracy in describing products and methods and the published permafrost map products are not consistent.

The training data set as it is described in this manuscript using a large data collection of MAGT at ZAA lacks transparency and is not publicly available. The major revision requirements are better product descriptions and higher transparency on the training data collection on MAGT in ZAA as most important issue.

These are the main points of concern that should be solved by providing more details and discussion on MAGT in ZAA:

i) The depth of ZAA is stongly changing throughout the Northern hemisphere: e.g. at higher latitudes minimum and maximum air temperature span a much large temperature range than at mid latitudes. In case of this large temperature range the ZAA depth is only reached at deeper ground depths of 10 to 15 m. This is in contrast to ZAA at more shallower depths in discontinuous permafrost and mid latitude regions. This means the depth of MAGT varies considerably in this map product, please add this to discussion, Is it possible to add the depth of ZAA as an additional metadata raster in the product? Please expand on this in the discussion chapter. This is also relevant for comparison with other products because mapped regional, circumarctic, global MAGT products and simulations in other communities refer to MAGT in specific depths always.

ii) As the authors state the MAGT at ZAA training data is based on the most comprehensive field data collection by Alto et al. 2018. However, this higher level data collection derived from various sources is not publicly available. Alto et al. 2018 describe in their comprehensive manuscript in detail the methods and the sources of the data. The authors describe how for extracting MAGT at ZAA or close to ZAA they manually calculated these data from ground temperature depth profiles from the different data providers (GTN-P data base, Roshydromet, national PIs). However, in context of this MAGT at ZAA map product there are open questions: for example Roshydromet temperature depth profiles have a standardized maximum depth of 3.20 m. What value exactly represents MAGT at ZAA in regions with considerably deeper ZAA depths then 3 meters? Please show transparency on this issue and discuss. Please also provide the detail on how you averaged the different temporal resolutions (e.g., hourly, daily etc measurements) of the ground temperature input data sets.

iii) The majority of the MAGT sites of this data collection are not within permafrost zones (continuous, discontinuous, isolated) and do not represent 'permafrost' temperatures. Please show the share of 'permafrost' vs non permafrost MAGT at ZAA training data and could you add an estimate of different accuracies in deriving 'permafrost' vs non permafrost MAGT, at least the authors should make readers aware of this issue and discuss it.

iv) The MAGT at ZAA data collection in Alto et al. 2018 refers to the time span 2000 – 2014. The presented state of permafrost in the raster layer is from 2010 to 2016. Eventually the authors have explained the temporal representativeness of the training data set related to the time span from 2010 to 2016 in their manuscript. If the authors did they should describe it more clearly, if not the authors should add this information.

This referee comments do not implement that the produced raster sets are not valid – they are of value and should be used in several communities - but the accuracy of these products stated in this manuscript is unrealistic already by the nature and the noise of the MAGT at ZAA input data.

In summary, an additional raster or other form of meta data information on the depth of MAGT is required for a good usage of the mapped permafrost products in other, also permafrost-not experienced communities: discussions and details on  i) to iv) should be provided in the manuscript.

The other input data should be described more clearly, stating data sources, exact product names, native spatial resolution, temporal resolution and time stamps of the products, e.g. also in the form of a table.

Examples are the source of the lake data set is unclear, its native spatial resolution, also the sentence 'small lakes were filtered out by majority statistical processes' remains unclear. Still other large surface water bodies, such as the large Arctic rivers are not excluded. This data treatment does not seem to be consistent. Please discuss.

When showing the permafrost extent of the Northern hemisphere, could the authors add also the values including the lake area for a comparison with other permafrost map products that have lake areas included?

The inspection of the published map products Ran et al. 2021 https://data.tpdc.ac.cn/en/data/5093d9ff-a5fc-4f10-a53f-c01e7b781368/ shows that large lakes are not excluded, e.g. the area of the deep Lake Baikal in Siberia contains MAGT at ZAA values. The authors need to correct their product masking surface waters and upload a new version.

[Figure]

The figure shows a snapshot of the MAGT product with the area of Lake Baikal containing MAGT patterns looking similar to bathymetry related features?

It would be user-friendly to convert the GIS no data value of – 9999 into a more user-friendly no data value, e.g. NaN.

---

## Referee Report (RR2)

**Review comments on ESSD-2021-83**

Yohua Ran et al. present a relevant raster data collection of permafrost related ground quantities, these are the GCOS Essential Climate Variables ECVs: Mean Annual Ground Temperature (MAGT) that the authors refer to at the Zero Annual Amplitude (ZAA), and Active Layer Thickness (ALT) and permafrost probability.

Still there is the mismatch of the MAGT training data set that includes a wide range of non-ZAA measurement depths, e.g. Roshydromet boreholes reach a maximum depth of 3.20 m only, not ZAA.

The MAGT raster product represents mapped MAGT in a ground depth range of around 3 m to 25 m, potential users of the MAGT product Ran et al. should be made aware of this fact, specifically if they want to undertake comparisons with their own ground temperature data or with other map products. The authors should make this statement in the abstract, in the manuscript text, in the figure captions when the MAGT product is shown and also in the abstract of the data publication landing site.

In their edited new version, the authors added more information on the MAGT training data and the reference of Karjalainen et al. 2019 and added a paragraph discussing the variation in ZAA. It is understandable that from this multitude of contributors and programs no data publication of the ground data can be required. However, transparency is still lacking, specifically information on the range of the depth of the MAGT value. In this context, it is inevitable that the authors need to show an overview – in the form of a table on the content of the MAGT training data covering the time window 2000-2016

This information is needed stratified related to program/source/author contributing the MAGT values: program/source/author for the time window 2000-2016

i) the authors should indicate the number n of stations that they used from the respective sources, ii) important: the authors need also to indicate the potential depth range min, max – or in case of Roshydromet the chosen depth of 3.20 m iii) the authors should indicate the estimated percentage of known ZAA depths to this group (the authors provide in the text an estimation of ca 75% for the full data set), e.g. in case of Roshydromet it would be zero.

The authors should add following discussion points

i) the authors need to include a discussion and a reference on the most actual mapped permafrost GCOS ECV products that are available: the European Space Agency ESA provide in their Climate Change Initiative CCI program also GCOS ECV Permafrost products. The first version of Permafrost CCI MAGT, ALT and permafrost probability products were released already in 2020, these products were already available when Ran et al. composed this manuscript. Since spring 2021, the newest version of Permafrost CCI MAGT, ALT and permafrost probability products are available for download to the permafrost and climate science communities.

ii) the authors should add a paragraph in discussion, on the fact if they are comparing the spatial extent of products, often in discontinuous zones permafrost still may occur at deeper layer but has thawed in lower depths. E.g. if the authors compare permafrost probability a product with a low specific product depth, e.g. 2 m, might contain no permafrost, but the Ran et al. product might contain permafrost because it relates to a deeper ground depth.

iii) the authors should be more careful in relation to their ALT map product: Training data are rare for the vast Siberian region. E.g. comparing the Ran et al. product with existing ALT and Active Layer

Depth measurements in Siberia the ALT product shows unreliable ALT data for the lower latitudes: they are far to low for large parts of Yakutia. In contrast, in several Siberian mountain regions ALT (ranges in the Ran et al. product > 1 m) seem to be overestimated by a magnitude of 3 to 5. The authors should discuss carefully these regional gaps in the training data set.

Data publication:

There is no read me file or product description available in the downloaded data package. The authors should add a Read me file

The authors should provide information on the Units for ALT, MAGT, permafrost probability

Also information on no data value is required to enable easy re-use of the product.

Would be good to provide in the read me file the information on the spatial resolution and also in the abstract text of the landing page.

---

## Author Response (AR2)

**Responses to Reviewer's Comments and Suggestions**
**(Responses are shown in blue.)**

**Review comments on ESSD-2021-83**

Youhua Ran et al. present a relevant raster data collection of permafrost-related ground quantities, these are the GCOS Essential Climate Variables, ECVs: mean annual ground temperature MAGT at zero annual amplitude (ZAA), and active layer thickness (ALT) representative for a thermal state of permafrost for the time window from 2010 to 2016. In addition, the authors also derived permafrost probability and what is the novelty: aridity-index related permafrost regions.

Despite the high value of providing these mapped permafrost related quantities and the very interesting novel approach integrating the aridity, the manuscript still lacks clarity and accuracy in describing products and methods and the published permafrost map products are not consistent.

The training data set as it is described in this manuscript using a large data collection of MAGT at ZAA lacks transparency and is not publicly available. The major revision requirements are better product descriptions and higher transparency on the training data collection on MAGT in ZAA as most important issue.

Response: Thank you. We have carefully revised the paper to provide better product descriptions and higher transparency regarding the collection of the training data.

These are the main points of concern that should be solved by providing more details and discussion on MAGT in ZAA:

i) The depth of ZAA is stongly changing throughout the Northern hemisphere: e.g. at higher latitudes minimum and maximum air temperature span a much large temperature range than at mid latitudes. In case of this large temperature range the ZAA depth is only reached at deeper ground depths of 10 to 15 m. This is in contrast to ZAA at more shallower depths in discontinuous permafrost and mid latitude regions. This means the depth of MAGT varies considerably in this map product, please add this to discussion, Is it possible to add the depth of ZAA as an additional metadata raster in the product? Please expand on this in the discussion chapter. This is also relevant for comparison with other products because mapped regional, circumarctic, global MAGT products and simulations in other communities refer to MAGT in specific depths always.

Response: Yes, the depth of MAGT varies for this MAGT product. Your idea to add the depth of ZAA as an additional raster is a good one. However, the exact depth of ZAA is not available for specific boreholes. To increase transparency, a more detailed description of the MAGT measurement data has been added in section 2.1. Furthermore, a paragraph discussing the variation of ZAA depth has been added.

ii) As the authors state the MAGT at ZAA training data is based on the most comprehensive field data collection by Alto et al. 2018. However, this higher level data collection derived from various sources is not publicly available. Alto et al. 2018 describe in their comprehensive manuscript in detail the methods and the sources of the data. The authors describe how for extracting MAGT at ZAA or close to ZAA they manually calculated these data from ground temperature depth profiles from the different data providers (GTN-P data base, Roshydromet, national PIs). However, in context of this MAGT at ZAA map product there are open questions: for example Roshydromet

temperature depth profiles have a standardized maximum depth of 3.20 m. What value exactly represents MAGT at ZAA in regions with considerably deeper ZAA depths then 3 meters? Please show transparency on this issue and discuss. Please also provide the detail on how you averaged the different temporal resolutions (e.g., hourly, daily etc measurements) of the ground temperature input data sets.

Response: Thank you. We have provided a more detailed description of the MAGT measurement data in section 2.1. A citation, i.e., Karjalainen et al., 2019, has been added to improve the transparency regarding the MAGT data.

Regarding your question, "What value exactly represents MAGT at ZAA in regions with considerably deeper ZAA depths then 3 meters?", we do not know the exact MAGT at ZAA in regions where we only have values to a depth of 3.20 m (and where the actual ZAA depth (DZAA) is greater). However, in light of the discussion in Karjalainen et al (2019), we assume that the MAGT at 3.20 m based on a year-round time series is representative of the MAGT at the ZAA.

Regarding the averaging ground temperature values, in the case of the RosHydromet data, it is true that the measurements were not necessarily from the DZAA, and we averaged the temperature time series at 3.20 m depth to compute MAGT. In this study, we ensured that we included only those time series that covered entire years, that is, all 12 months or 365 days (although in some cases, several days were missing; however, this was considered unlikely to greatly affect the annual averages). All provided temperature values were used in these computations apart from those values flagged unreliable by the data providers.

These techniques apply to all of the used time series input data. When the ground temperatures at the DZAA were in question (and intraannual variability was thus minimal), some interannual temperature trends were usually visible (oftentimes increasing), and all available full years (or months, days or hours) from the 2000-2014 period were used to compute MAGT.

iii) The majority of the MAGT sites of this data collection are not within permafrost zones (continuous, discontinuous, isolated) and do not represent 'permafrost' temperatures. Please show the share of 'permafrost' vs non permafrost MAGT at ZAA training data and could you add an estimate of different accuracies in deriving 'permafrost' vs non permafrost MAGT, at least the authors should make readers aware of this issue and discuss it.

Response: This is a good point. We have added "The accuracy of the predicted MAGT in permafrost regions, with field measurements in permafrost sites (MAGT $\leqslant 0$ ℃) used as reference, was significantly higher (RMSE=1.06 ℃, bias=-0.22 ℃) than that in nonpermafrost regions (RMSE=1.56 ℃, bias=0.88 ℃" to the revised manuscript.

iv) The MAGT at ZAA data collection in Alto et al. 2018 refers to the time span 2000 – 2014. The presented state of permafrost in the raster layer is from 2010 to 2016. Eventually the authors have explained the temporal representativeness of the training data set related to the time span from 2010 to 2016 in their manuscript. If the authors did they should describe it more clearly, if not the authors should add this information.

Response: The collection of the MAGT at ZAA data in this study corresponds to the time span

2000-2016. The presented thermal state of permafrost in the raster layer corresponds to the period from 2000 to 2016.

This referee comments do not implement that the produced raster sets are not valid – they are of value and should be used in several communities - but the accuracy of these products stated in this manuscript is unrealistic already by the nature and the noise of the MAGT at ZAA input data.

Response: Yes, permafrost is more of a climate product; it is also a product of ecosystems in some cases. According to the change characteristics of permafrost and the current level of prediction accuracy, the data released in this study represent the thermal state of permafrost at the Northern Hemisphere scale for the period 2000-2016.

In summary, an additional raster or other form of meta data information on the depth of MAGT is required for a good usage of the mapped permafrost products in other, also permafrost-not experienced communities: discussions and details on i) to iv) should be provided in the manuscript.

The other input data should be described more clearly, stating data sources, exact product names, native spatial resolution, temporal resolution and time stamps of the products, e.g. also in the form of a table.

Response: This is a good idea. Table 1 summarizes the environmental and climate variable datasets used in this study to predict MAGT and ALT.

**Table 1: Environmental and climate variable datasets used in this study to predict MAGT (mean annual ground temperature) and ALT (active layer thickness). MODIS, Moderate Resolution Imaging Spectroradiometer; LST, land surface temperature; AVHRR, Advanced Very High Resolution Radiometer; GLASS, Global Land Surface Satellite).**

| Variable | Data source | Spatial resolution | Temporal resolution and time span |
|---|---|---|---|
| Freezing degree-days, ℃-days | MODIS LST | 1 km | Daily, 2000-2016 |
| Thawing degree-days, ℃-days | MODIS LST | 1 km | Daily, 2000-2016 |
| Snow cover duration, days | MODIS, AVHRR | 0.05 ° | Half-month, 2000-2016 |
| Leaf area index | GLASS | 1 km | Eight-day, 2000-2016 |
| Precipitation, mm | WorldClim v2.1 | 1 km | 1970–2000 but adjusted to 2000–2016 |
| Solar radiation, kJ $m^{-2}$ $day^{-1}$ | WorldClim v2.1 | 1 km | 1970–2000 but adjusted to 2000–2016 |
| Soil organic content, g $kg^{-1}$ | SoilGrids250 | 250 m | - |
| Soil bulk density, kg $m^{-3}$ | SoilGrids250 | 250 m | - |
| Coarse fragment content, vol % | SoilGrids250 | 250 m | - |

Examples are the source of the lake data set is unclear, its native spatial resolution, also the sentence 'small lakes were filtered out by majority statistical processes' remains unclear. Still

other large surface water bodies, such as the large Arctic rivers are not excluded. This data treatment does not seem to be consistent. Please discuss. When showing the permafrost extent of the Northern hemisphere, could the authors add also the values including the lake area for a comparison with other permafrost map products that have lake areas included?

The inspection of the published map products Ran et al. 2021 https://data.tpdc.ac.cn/en/data/5093d9ff-a5fc-4f10-a53f-c01e7b781368/ shows that large lakes are not excluded, e.g. the area of the deep Lake Baikal in Siberia contains MAGT at ZAA values in spatial patterns related to the bathymetry of Lake Baikal. The authors need to correct their product masking surface waters and upload a new version.

It would be user-friendly to convert the GIS no data value of – 9999 into a more user-friendly no data value, e.g. NaN.

Response: Thank you. We have clarified the description of the lake data used in this study. The extent of lakes was sourced from the global lakes and wetlands database, level 1 (Lehner and Döll, 2004), which comprises large lakes (area $\geq$ 50 km$^2$) and large reservoirs (storage capacity $\geq$ 0.5 km$^3$). All of the results have been updated in the revised manuscript, and new data products have been uploaded.

---

## Author Response (AR3)

**Comments to the author:**

I have received the review report with significant change requests. Please address them in your revised version of the manuscript.

**Review comments on ESSD-2021-83**

Yohua Ran et al. present a relevant raster data collection of permafrost related ground quantities, these are the GCOS Essential Climate Variables ECVs: Mean Annual Ground Temperature (MAGT) that the authors refer to at the Zero Annual Amplitude (ZAA), and Active Layer Thickness (ALT) and permafrost probability.

Still there is the mismatch of the MAGT training data set that includes a wide range of non-ZAA measurement depths, e.g. Roshydromet boreholes reach a maximum depth of 3.20 m only, not ZAA.

The MAGT raster product represents mapped MAGT in a ground depth range of around 3 m to 25 m, potential users of the MAGT product Yohua Ran et al. should be made aware of it, specifically if they want to undertake comparisons with their own ground temperature data or with other map products. The authors should make this statement in the abstract, in the manuscript text, in the figure captions when the MAGT product is shown and also in the abstract of the data publication landing site.

Response: Thank you for your suggestion. We have added this statement to the abstract, the manuscript text (Sections 2.1 and 3.1), the caption of Figure 3 and the abstract of the data site.

In their edited new version, the authors added more information on the MAGT training data and the reference of Karjalainen et al. 2019 and added a paragraph discussing the variation in ZAA. It is understandable that from this multitude of contributors and programs no data publication of the ground data can be required. However, transparency is still lacking, specifically information on the range of the depth of the MAGT value. In this context, it is inevitable that the authors need to show an overview – in the form of a table on the content of the MAGT training data covering the time window 2000-2016

This information is needed stratified related to program/source/author contributing the MAGT values: program/source/author for the time window 2000-2016

i) the authors should indicate the number n of stations that they used from the respective sources, ii) important: the authors need also to indicate the potential depth range min, max – or in case of Roshydromet the chosen depth of 3.20 m iii) the authors should indicate the estimated percentage of known ZAA depths to this group (the authors provide in the text an estimation of ca 75% for the full data set), e.g. in case of Roshydromet it would be zero.

Response: According to your helpful suggestion, we have provided an overview of MAGT data in the form of a table (Supplementary Table 1). "For the significant portion of boreholes (>75%)

without confirmed depth of ZAA" is a clerical error. This means that the depths of 75% of the sites are deeper than 8 m. We have removed this statement and improved the description.

The authors should add following discussion points
i) the authors need to include a discussion and a reference on the most actual mapped permafrost GCOS ECV products that are available: the European Space Agency ESA provide in their Climate Change Initiative CCI program also GCOS ECV Permafrost products. The first version of Permafrost CCI MAGT, ALT and permafrost probability products were released already in 2020, these products were already available when Ran et al. composed this manuscript. Since spring 2021, the newest version of Permafrost CCI MAGT, ALT and permafrost probability products are available for download to the permafrost and climate science communities.

Response: Thank you. This product is surprising and provided an annual ALT and ground temperature at various depths during 1997-2019. We have added this in section 1. A citation was also added.

"The European Space Agency (ESA) Climate Change Initiative (CCI) also provide permafrost products including MAGT, ALT and permafrost probability that are derived from a remote sensing-driven CryoGrid model (Obu et al., 2021). The annual ALT and ground temperature at various depths during 1997-2019 with 1-km resolution are uniquely available for the permafrost and climate science communities, although broad validation is needed, especially for ALT products."

ii) the authors should add a paragraph in discussion, on the fact if they are comparing the spatial extent of products, often in discontinuous zones permafrost still may occur at deeper layer but has thawed in lower depths. E.g. if the authors compare permafrost probability a product with a low specific product depth, e.g. 2 m, might contain no permafrost, but the Ran et al. product might contain permafrost because it relates to a deeper ground depth.

Response: Thank you. Our comparison confirms your comments. A short paragraph has been added in section 3.2 to clarify this issue, as follows:

"This result is generally consistent with those of existing studies (Zhang et al., 2008; Gruber, 2012; Chadburn et al., 2017; Aalto et al., 2018; Obu et al., 2019) and logically consistent with the ESA CCI permafrost area defined according to MAGT at 2 m (Obu et al., 2021). Our results contain permafrost in discontinuous zones while ESA CCI data contain no permafrost, because permafrost still may occur in deeper layers but has thawed at shallower depths."

iii) the authors should be more careful in relation to their ALT map product: Training data are rare for the vast Siberian region. E.g. comparing the Ran et al. product with existing ALT and Active Layer Depth measurements in Siberia the ALT product shows unreliable ALT data for the lower latitudes: they are far to low for large parts of Yakutia. In contrast, in several Siberian mountain regions ALT (ranges in the Ran et al. product > 1 m) seem to be overestimated by a magnitude of 3 to 5. The authors should discuss carefully these regional gaps in the training data set.

Response: Thank you. Yes, the uncertainty of the predicted ALT is considerable, which is strongly limited by the availability and spatial representation of training data. According to your very enlightening comments, we have added this passage in Section 3.1.

"Of course, the uncertainty of ALT is considerable, especially in the vast area of western Siberia

where the training data are sparse. The low spatial representativeness of training data may lead to an overestimation in several Siberian mountain regions and underestimation near the lower boundary of permafrost. This highlights the importance and urgency of strengthening global coordinated ALT observation networks."

Data publication:

There is no read me file or product description available in the downloaded data package. The authors need to add a Read me file

They should provide information on the Units for ALT, MAGT, permafrost probability

Also information on no data value is required to enable easy re-use of the product.

Would be good to provide in the read me file the information on the spatial resolution and also in the abstract text of the landing page.

Response: Thank you. These are good suggestions. We have added a short readme file with the data package. The units and spatial resolution of the datasets are provided in the read me file. The abstract of the data landing page will also be updated. Moreover, the ESSD data paper, which the best product description, will be available in the data package. Because it is only several files, not long time series data, we prioritize supporting software reading and keep the "no data" value (i.e., -9999) that is compatible with ArcGIS.

---

## Author Response (AR4)

**Responses to Reviewer's Comments and Suggestions**
**(Responses are shown in blue.)**

**Review comments on ESSD-2021-83**

Yohua Ran et al. present a raster data collection of permafrost related ground quantities, these are the GCOS Essential Climate Variables ECVs: Mean Annual Ground Temperature (MAGT) that the authors refer to at the Zero Annual Amplitude (ZAA), and Active Layer Thickness (ALT) and permafrost probability and added the detailed information on the training and validation ground data, the sources, and the depth range of the measurements.

The detailed table adequately provides the overview on the ground data that were used as training data, the table provides the sources, the depth ranges of the measurements, number of input data and the time window.

In their edited new version, the authors added this information as supplement. I strongly recommend to integrate this table as table text within the manuscript itself, e.g. within the appendix, as it is of high informative value and may be underused by the readers if it will be provided as supplement only.

A further minor correction. The authors added new text "The European Space Agency (ESA) Climate Change Initiative (CCI) also provide permafrost products including MAGT, ALT and permafrost probability that are derived from a remote sensing-driven CryoGrid model (Obu et al., 2021). The annual ALT and ground temperature at various depths during 1997-2019 with 1-km resolution are uniquely available for the permafrost and climate science communities, although broad validation is needed, especially for ALT products." please change the statement 'although broad validation is needed' accordingly as the validation is always an official part of ESA CCI programs and validation reports on all ESA CCI products are regularly updated and available, this means this also applies to the ESA CCI permafrost products.

Response: Thank you. We have added the table in the appendix according to your suggestion. The statement 'although broad validation is needed, especially for ALT products.' was changed also.